# Human orbitofrontal cortex signals decision outcomes to sensory cortex during behavioral adaptations

Bin A. Wang [1,2], Maike Veismann [1,2], Abhishek Banerjee [3] ✉ & Burkhard Pleger [1,2] ✉

The ability to respond flexibly to an ever-changing environment relies on the orbitofrontal cortex (OFC). However, how the OFC associates sensory information with predicted outcomes to enable flexible sensory learning in humans remains elusive. Here, we combine a probabilistic tactile reversal learning task with functional magnetic resonance imaging (fMRI) to investigate how lateral OFC (lOFC) interacts with the primary somatosensory cortex (S1) to guide flexible tactile learning in humans. fMRI results reveal that lOFC and S1 exhibit distinct task-dependent engagement: while the lOFC responds transiently to unexpected outcomes immediately following reversals, S1 is persistently engaged during re-learning. Unlike the contralateral stimulus-selective S1, activity in ipsilateral S1 mirrors the outcomes of behavior during re-learning, closely related to top-down signals from lOFC. These findings suggest that lOFC contributes to teaching signals to dynamically update representations in sensory areas, which implement computations critical for adaptive behavior.

Humans and animals learn to rapidly adapt their behavior to new environmental challenges, which is critical for survival[1]. The flexibility in adjusting the decision strategy, based on the prediction and evaluation of behavioral outcomes, is a prerequisite for adaptive behavior and is severely compromised in many psychiatric disorders[2]. Among the elaborate frontal cortical areas involved in flexible decision-making, the orbitofrontal cortex (OFC) has been one of the most intensively studied structures and is known to have widespread connectivity to sensory areas, as well as to cortical and subcortical areas related to memory, learning and attention[3,4]. OFC is specifically implicated in choosing objects or an action based on the expected outcome value and updating the value of different stimulus-outcome associations[5,6]. Compared to medial OFC, which encodes the reward value to support choices, lateral OFC (lOFC) is relatively more specialized for assigning credit for both positive and negative outcomes to specific stimulus choices, emphasizing the lOFC's role in learning the values of options[7]. We recently identified the lOFC in humans as an important brain region related to updating the decision strategy based on newly accumulated

evidence[8]. In this context, lOFC is encoding the prediction error (PE) in the face of environmental changes, thereby updating associative representations in other brain areas and, ultimately, guiding adaptive behavior[9].

Associating sensory stimuli with predicted outcomes is essential for successful learning and adaptive behavior. One way in which the brain might perform this operation is by conveying a 'teaching' signal, based on choice outcomes, to sensory areas involved in stimulus processing[10–12]. Several studies have provided evidence consistent with this assumption, showing responses in primary sensory cortices related to the expectation of a stimulus or reward[13–16], which top-down signals from OFC may mediate. Studies in rodents have uncovered the distinct rules of how OFC exerts 'teaching' signals to modulate sensory processing[17,18]. Recently, using a tactile reversal learning task in rodents, Banerjee et al. revealed that the top-down signal from lOFC updated sensory representations in the primary somatosensory cortex (S1) by remapping responses of a subpopulation of value neurons sensitive to reward history[19]. In humans, it remains unclear whether

[1]Department of Neurology, BG University Hospital Bergmannsheil, Ruhr-University Bochum, Bochum, Germany. [2]Collaborative Research Centre 874 "Integration and Representation of Sensory Processes", Ruhr-University Bochum, Bochum, Germany. [3]Adaptive Decisions Lab, Biosciences Institute, Newcastle University, Newcastle upon Tyne, UK. ✉e-mail: abhi.banerjee@newcastle.ac.uk; Burkhard.V.Pleger@ruhr-uni-bochum.de

comparable top-down signals from the lOFC instruct sensory areas to remap stimulus-outcome associations essential for behavioral adaptation. If validated, these observations may crucially reveal common cross-species circuit motifs underlying learning and flexibility within the same sensory domain.

In the current study, we implement a probabilistic Go/NoGo reversal learning task and deploy functional magnetic resonance imaging (fMRI) to record human brain activity. To decipher human behavior in a probabilistic environment, we combine computational models of behavior and fMRI data analyses, including univariate and multivariate analyses, to synergize insights about task-dependent neural computations of behavioral adaptation in humans. We show that human participants learn the task-relevant conditional probabilities of stimuli and dynamically update their learning rate accordingly. Additionally, fMRI analysis reveals that the prediction error-related activity in lOFC responds transiently to unexpected reward and non-reward outcomes after reversals and decreases as participants re-learned the task. In contrast, S1 neural activity reflects initial learning of stimulus-response associations and later engagement upon re-learning phase. By leveraging multivariate representational similarity analysis (RSA) on fMRI data, we reveal that activity in lOFC represents the choice outcomes after the reversal and during RE. In contrast, activity in the contralateral S1 represents the sensory stimulus. Ipsilateral S1, in turn, reflects the choice outcomes during RE, which is related to top-down signals from lOFC. These findings show that flexible decision-making in humans relies on comparable computational foundations in lOFC and S1, as reported in mice[19].

## Results

### Experimental design

We designed a probabilistic Go/NoGo reversal learning task for humans in which the associations between two tactile stimuli and responses are initially learned over a series of trials and then reversed (Fig. 1a–c). Participants had to ascertain which response ('Go' or 'NoGo') to each tactile stimulus was the best to obtain a reward by trial and error. In each block, two new tactile patterns were randomly selected from the eight alternative patterns shown in Fig. 1b. One of the two responses for each tactile cue had a higher reward probability than the other ($p = 0.7$ versus $p = 0.3$, Fig. 1c). Within each block, we switched the associations between stimuli and responses at a random trial dividing the block into two phases: (1) the initial learning phase, in which the participants learned the stimulus-response association for each stimulus, and (2) the reversal phase, in which they had to reverse their choice preference to maximize the received reward (Fig. 1c).

### Humans learned the probability distribution and dynamically updated their learning rate

First, we analyzed the performance during both the initial learning and re-learning phases after the reversal of stimulus-response associations. We aligned the reversal phase of each block using the reversal point and averaged the proportion of correct responses across blocks. At the beginning of the block, participants quickly learned the stimulus-response association. After the stimulus-response association was switched, the performance dramatically dropped and gradually increased again while participants reversed their choice behavior (Fig. 1d). To investigate the dynamic changes along the learning process, we subdivided task performance into 'learning naïve' (LN) and 'learning expert' (LE) in the initial learning phase, and 'reversal naïve' (RN), 'reversal expert' (RE) in the reversal phase. Based on the group performance, we selected the first ten trials in both training periods, pre- and post-reversal, as LN and RN, respectively, and the last ten trials immediately before the reversal or task completion as LE and RE, respectively. For fMRI analyses, we only considered these respective trials. We compared the fraction of correct responses between the expert and naïve periods and found a significantly higher proportion

of correct responses in the expert period for both the initial learning and the reversal learning phase (initial learning, $t_{(31)} = 9.04$, $p = 7.54 \times 10^{-9}$; reversal learning, $t_{(31)} = 20.43$, $p = 6.98 \times 10^{-19}$, Fig. 1e).

To test how participants switched their decision strategy based on the previous decision outcome, we applied four different computational models (M1: Random Responding; M2: Win-Stay-Lose-Switch (WSLS); M3: Rescorla–Wagner (RW) and M4: Hierarchical Gaussian Filter (HGF). For more details about these four models please refer to the 'Methods' section). First, we simulated the responses of the four models given a set of particular parameters which were independent of the participant's actual behavioral responses. We then tested one measure that captured fundamental aspects of the flexible decision-making process based on prior experience: the probability of repeating a decision, $p$(staying). Based on the model simulations across a range of parameter settings (Supplementary Fig. 1), we chose a particular set of parameters for the response module of the four models (M1: $b = 0.5$; M2: $\varepsilon = 0.05$; M3: $\beta = 5$; M4: $\zeta = 0.5$). For the free parameters of the additional perceptual module in RW ($\alpha$) and HGF ($\omega$), we utilized the 'Bayes optimal' values for our experimental input sequence based on a free energy minimization approach (Supplementary Table 1). These 'Bayes optimal' values are independent of the participant's actual behavioral responses. For more details about these model simulations, please refer to the 'Methods' section. Based on these parameters, we finally simulated the responses and calculated $p$(staying) for the four models. To compare the simulated responses with the observed responses, we also plotted $p$(staying) of participants' actual behavioral data. The WSLS model exhibited a significant dependence on past decision outcomes, whereas the Random Responding model showed no such dependence (Fig. 1f). Compared to these two models, participants' actual performance was better captured by both the RW and the HGF models (Fig. 1f), suggesting that participants' decisions were more likely determined by the updated choice value than random responding or past outcomes.

Next, we questioned whether participants' learning behavior could be rather explained by hierarchical learning (i.e., Bayesian HGF model), which includes dynamic updates of the learning rate based on individual learning trajectories, or by a fixed 'ideal' learning rate as assumed by the reinforcement learning algorithm (RW). By fitting the models to the participants' actual behavioral responses, the maximum a posteriori estimates of the free parameters (Supplementary Table 2) and the log-model evidence (LME) as the negative variational free energy under the Laplace assumption for both the RW and HGF model were assessed. To identify the model that best explained participants' behavior, we applied the random-effect Bayesian model selection (BMS), which assesses the relative plausibility of competing models based on LME. At the group level, BMS revealed posterior model probabilities of 95% for the winning HGF model (posterior probabilities: 0.95; exceedance probability = 1.00, Fig. 1g). Furthermore, we compared relative LME between the HGF and RW models at the individual level and showed that HGF was superior in 28 out of 32 participants (Fig. 1h). In addition to the LME, we calculated the Bayesian information criterion (BIC), which confirmed that the HGF was superior to the RW model (Supplementary Fig. 2). To validate the winning HGF model, we additionally performed cross-validations by predicting the responses in held-out data (Supplementary Fig. 3) and ensured that the HGF model successfully captured the real behavior of the whole experiment using the optimized parameters from model fitting (Supplementary Fig. 4). These results provide evidence that the participants learned the task-relevant conditional probabilities of stimuli and dynamically updated their learning rate.

### Representation of outcome prediction errors in the lOFC

Next, we investigated whether violations of outcome expectation evoked prediction error (PE) signals in the lOFC. We used the winning HGF model to drive the subject-specific estimates of outcome

PE ($\delta_1^{(t)}$) at the second level. The HGF model contains subject-specific parameters optimized using the participants' actual behavioral responses and therefore allows for individual expression of (approximate) Bayes-optimal learning. The outcome PE is the difference between the actual outcome and its a priori probability (i.e., before response outcome observation) according to the model. The unsigned outcome PE (i.e., absolute value) was included in the GLM as a parametric modulator of finite impulse response function (FIR), time-locked to the onset of outcomes and regressed against the fMRI responses in each voxel. In line with previous studies in humans[20,21], the responses in lOFC were significantly correlated with outcome PE ($x = 40$, $y = 48$, $z = -2$, $t_{(31)} = 4.99$, $p = 0.036$, family-wise error (FWE) peak-level corrected for multiple comparisons using small-volume correction (SVC), Fig. 2). Additionally, we found a widely distributed set of cortical areas, including the middle frontal gyrus (MFG), the supplementary motor cortex (SMA), insular cortex (Ins) and posterior parietal cortex (PPC), that also correlated positively with the outcome PE, but only at an uncorrected threshold of $p < 0.001$ for the whole brain volume (Fig. 2, Supplementary Table 3).

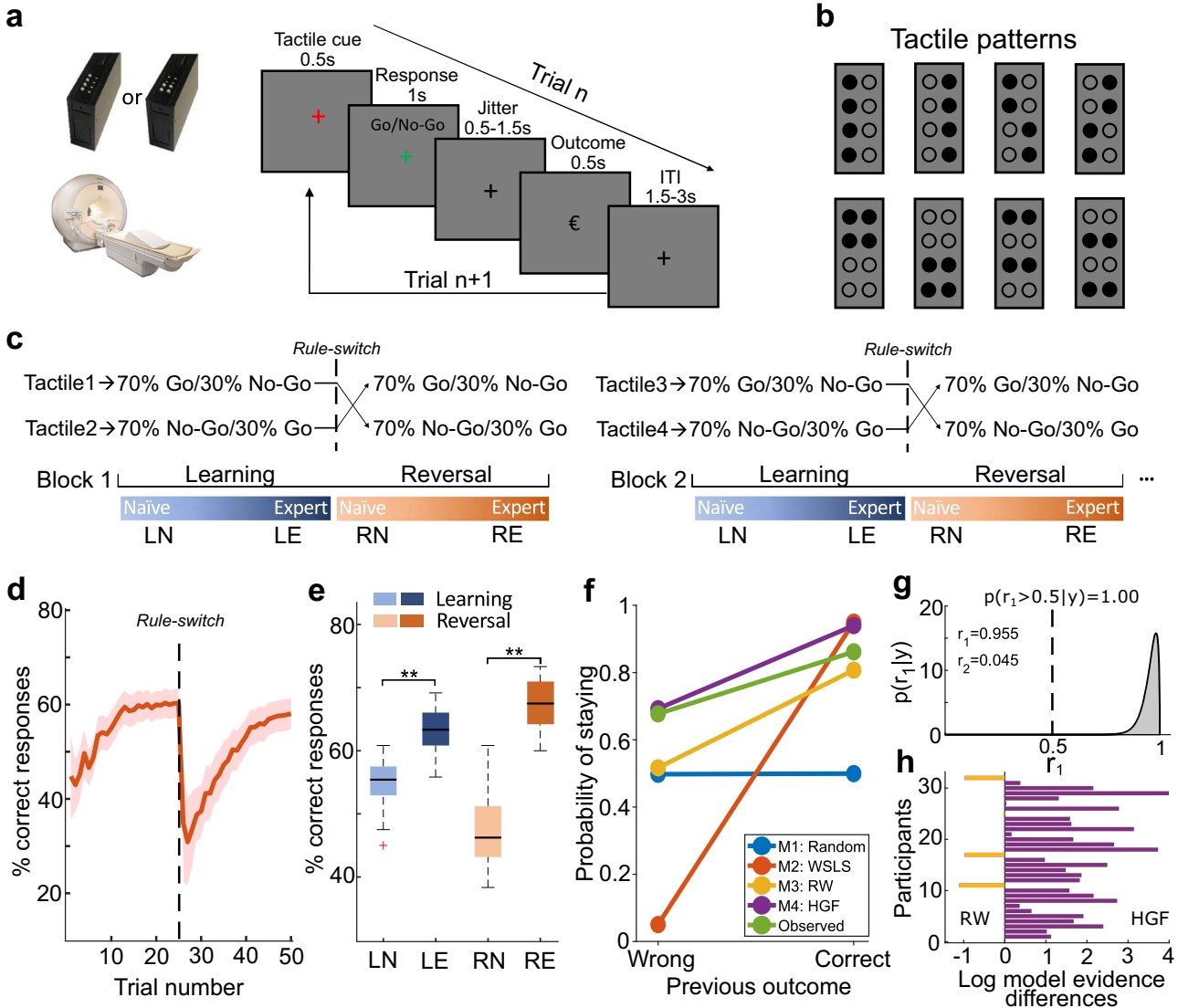

**Fig. 1 | Probabilistic Go/NoGo reversal learning task and behavioral performance in humans. a** Timeline of a single trial. **b** Eight tactile patterns were used for the task. **c** The illustration of the learning blocks. In each block, 70% of trials in which one of the two tactile patterns was presented were assigned to 'Go', whereas 70% of trials in which the alternative tactile pattern was presented were assigned to 'NoGo'. Within each block, the stimulus-response association was switched at a random trial (20–25). The trials were categorized into four different learning phases: 'learning naïve' (LN), 'learning expert' (LE), 'reversal naïve' (RN), and 'reversal expert' (RE). **d** The group averaged proportion of correct responses along with the learning process across blocks. The dashed line indicates the rule-switch. The red shaded area indicates the standard error of the mean (SEM, $n = 32$ participants). **e** The proportion of correct responses in each phase. The expert phase exhibited a significantly higher proportion of correct responses compared with the naïve phase

(Learning: $p = 7.54 \times 10^{-9}$; Reversal: $p = 6.98 \times 10^{-19}$; paired two-sample $t$-tests with two tails, $n = 32$ participants). Box plots indicate the median (middle line), 25th, and 75th percentile (box), and the maximum and minimum (whiskers) as well as the outlier (red cross). The asterisks indicate $p < 0.001$. **f** The probability of repeating a decision, $p$(staying), as a function of the outcomes of the previous trial for the simulated responses of four models (Random Responding, Win-Stay-Lose-Switch (WSLS), Rescorla–Wagner (RW) and Hierarchical Gaussian Filter (HGF)) and observed responses. **g** Dirichlet density describes the probability of the HGF model given the observed data across the group. The shaded area represents the exceedance probability. $r_1$, $r_2$ = conditional expectations of the probabilities of the HGF and RW model, respectively. **h** The individual difference of log-model evidence between the HGF and the RW model with positive values indexing participants who preferred the HGF model. Source data are provided as a Source Data file.

**Fig. 2 | fMRI activity related to the outcome PE.** Whole-brain analysis of correlations with outcome PE revealed responses in bilateral middle frontal gyrus (MFG), supplementary motor cortex (SMA), bilateral insular cortex (Ins.), right posterior parietal cortex (PPC), and right lateral orbitofrontal cortex (lOFC). The activations (one-sided t-test, $p < 0.001$, uncorrected, for display purpose only) were superimposed on sagittal, coronal, and axial slices of a standard T1-weighted image from the Colin27 brain template implemented in MRIcron. Chris Rorden's MRIcron, all rights reserved. Coordinates next to each slice indicate their location in MNI space. Red-yellow coding indicates the t-scores of activation intensities.

## Engagement of lOFC immediately following reversals, but S1 during re-learning

We next studied the involvement of two a priori hypothesized brain areas engaged in the task: S1, which is important for tactile discrimination and sensory-outcome association learning, and the lOFC, which is engaged in the assignment of outcome value. Our hypothesis for the lOFC was based on a series of lesion studies in humans[22] and pharmacogenetic silencing and lesion experiments in rodents[19, 23,24] that together accumulated compelling evidence for a specific causal role of the lOFC in reversal learning. To examine whether activity in these two regions related to outcomes following reversals and RE, we applied two independent general linear models (GLM) analyses, time-locked to the onset of the outcome. First, by comparing LE and RN trials, we observed significantly enhanced BOLD signals in the right lOFC immediately after switching the stimulus-response association ($x = 44$, $y = 40$, $z = -14$, $t_{(31)} = 4.95$, $p_{FWE-SVC} = 0.035$, Fig. 3a and Supplementary Fig. 5). Second, by comparing LE and RE trials, we identified bilateral S1, which showed a significantly higher BOLD signal in the RE trials (left, $x = -50$, $y = -20$, $z = 48$, $t_{(31)} = 5.85$, $p_{FWE-SVC} = 0.002$; right, $x = 30$, $y = -32$, $z = 60$, $t_{(31)} = 5.54$, $p_{FWE-SVC} = 0.005$). We did not find a comparable effect in lOFC ($p_{FWE-SVC} > 0.05$, Fig. 3b). Notably, the bilateral S1 regions identified here were assigned to the Brodmann area 3b (S1_3b), based on the SPM Anatomy Toolbox[25,26]. These results suggest the existence of distinct neural engagement during reversal learning: while lOFC responds robustly and transiently to the reversal, S1 is engaged in the RE after reversal.

The BOLD activity relative to baseline across all behavioral phases revealed that the lOFC presented modest activity during LN but diminished responses in LE (Fig. 3c). During RN, we again found transient but large lOFC responses to unexpected outcomes, which decreased as participants re-learned the task during RE (Fig. 3c). To test the potentially differential influence of appetitive and aversive outcomes, we separately analyzed lOFC activity in rewarded (HIT, CR) and unrewarded (FA, MISS) trials for each of the four learning phases separately (Supplementary Fig. 6). LOFC responded to both, unexpected reward and non-reward trials immediately after the reversal (RN, within-subject repeated measures ANOVA: no significant interaction between phases and types of trials ($F_{(1,31)} = 0.95$, $p = 0.42$) and the main effect of types of trials ($F_{(1,31)} = 0.74$, $p = 0.53$); only a significant main effect of phases ($F_{(1,31)} = 17.87$, $p = 0.00019$)), suggesting that lOFC encodes deviations from expected outcome values after rule reversals to assign credit to specific stimulus-response associations, irrespective of whether they have been rewarded or not[7]. In the initial learning phase (LN > LE), lOFC also encoded deviations from expected outcomes, suggesting a supportive role of lOFC in the initial learning of probabilistic cue-outcome associations. Notably, responses in lOFC during LN were observed only in humans, not in mice. This finding can be interpreted in the context of the task design, which was probabilistic for humans but deterministic for mice (see Discussion for further details). Further mechanistic investigations in mice under probabilistic demands are required. S1, on the other hand, was engaged in initial stimulus-response association learning (LE), and this engagement persisted after the reversals and during RE (RN and RE) (Fig. 3c). Please note that we applied two different tactile pin indentations over the learning phase in each block and participants had to steadily direct their attention towards these stimulus changes for correct discriminations and decisions. That is why we did not observe any adaptation processes in the form of gradually decreasing S1 responses, which are well known to occur if the same stimulus is repetitively applied to the same skin location over a longer time period[27].

## Representation of the stimulus and outcome in lOFC and S1 following reversals

Univariate analysis of the fMRI data revealed that reversals elicited PE-related signals in the lOFC. In theory, PE signals are used to update stimulus-outcome associations. To further explore how the associative information is represented in lOFC and S1, we considered whether the response patterns in lOFC and S1 retained the learned association or updated to the new stimulus-outcome association following reversals: i.e., whether they were more selective for the stimulus or the outcome (Fig. 4a). To this end, we leveraged a multivariate pattern analysis (i.e., RSA) on the fMRI data at the time of outcome presentation. The rationale behind choosing the presentation of the outcome and not of the stimulus as the onset is that at the time point of outcome presentation, the lOFC should assign credit for unexpected outcomes to specific stimuli by signaling the outcome values to the sensory cortex. The sensory cortex, in turn, should remain stimulus-selective in the moment of outcome presentation but may become outcome-selective later in the task due to the ongoing feedback from lOFC. Please note that the stimulus-selectivity includes the associated Go/NoGo response (i.e., stimulus–response selectivity). For RSA, the representational dissimilarity matrices (RDMs) were constructed based on the predicted correlation distance for trials before and after reversal. The representation of stimulus-selectivity is reflected by the similarity of outcomes for the same tactile stimulus in the initial learning and reversal phase (i.e., Go-tactile$_{learning}$ = NoGo-tactile$_{reversal}$, or HIT$_{learning}$ = CR$_{reversal}$ in terms of outcomes). In contrast, the representation of outcome-selectivity is reflected by the similarity of the same outcomes in the initial learning and reversal phase (i.e., HIT$_{learning}$ = HIT$_{reversal}$). We asked whether lOFC and S1 displayed these properties during the outcome presentation.

Figure 4a schematically presents the two RDM models and the similarity of response patterns before versus after the reversal. To assess both, the immediate effect of the reversal and the response adaptation after RE, we applied each brain region (lOFC, S1_3b) to each

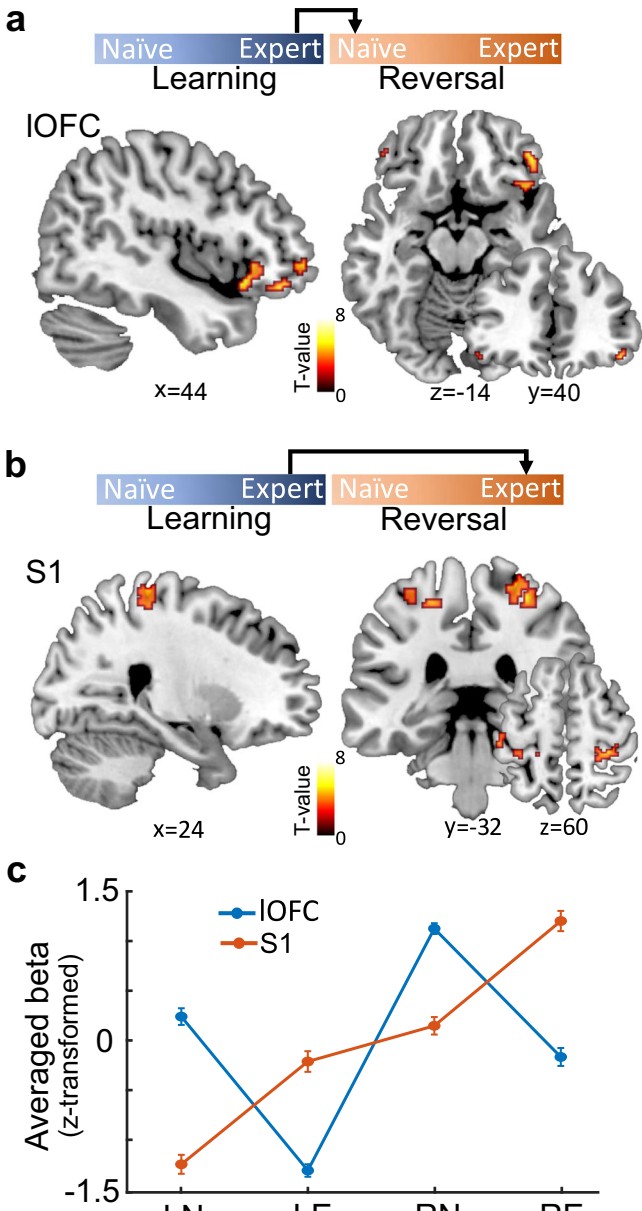

**Fig. 3 | Distinct engagement of lOFC and S1 after reversals and during re-learning, respectively. a** Significantly enhanced BOLD signals in lOFC immediately after the reversal (RN > LE, one-sided *t*-test, *p* = 0.035, family-wise error (FWE) peak-level corrected for multiple comparisons using small-volume correction (SVC)). **b** Significantly enhanced BOLD signals in bilateral S1 during re-learning after the reversal (RE > LE, one-sided *t*-test, left S1: *p* = 0.002, right S1: *p* = 0.005. FWE peak-level corrected for multiple comparisons using SVC). The activations (*p* < 0.001, uncorrected, for display purposes only) were superimposed on sagittal, coronal, and axial slices of a standard T1-weighted image from the Colin27 brain template implemented in MRIcron. Chris Rorden's MRIcron, all rights reserved. Coordinates next to each slice indicate their location in MNI space. Red-yellow coding indicates the *t*-scores of activation intensities. **c** The BOLD signals relative to baseline in lOFC and S1 across the four learning phases (LN, LE, RN, RE). The *y*-axis indicates the *z*-score of the mean beta value from lOFC and S1 as derived from the general linear model. The error bars indicate the SEM (*n* = 32 participants). Source data are provided as a Source Data file.

model twice: one analysis described the similarity of response pattern between LE and RN (LE → RN) and the other one between LE and RE (LE → → RE) (Fig. 4b). We found significant outcome-selective response pattern in lOFC after reversal (LE → RN, signed-rank test, $Z_{(31)}$ = 3.974,

*p* < 0.001; permutation test, effect size = 0.52, *p* = 0.021), and during RE (LE → → RE, signed-rank test, $Z_{(31)}$ = 4.628, *p* < 0.001; permutation test, effect size = 0.87, *p* = 0.0003, Fig. 4c). By contrast, response patterns in lOFC did not represent stimulus-selectivity, neither immediately after a reversal nor during RE (Supplementary Fig. 7). However, the response pattern in S1 was selective for the same tactile pattern (Go-tactile pattern or NoGo-tactile pattern) after RE as during initial learning (LE → → RE, signed-rank test, $Z_{(31)}$ = 1.861, *p* = 0.031), also when comparing the group mean against a null distribution generated by permuting the identity of trials in the RDM (permutation test, effect size = 0.59, *p* = 0.011, Fig. 4d). Interestingly, the response pattern in S1 during RE was outcome-selective (LE → → RE, signed-rank test, $Z_{(31)}$ = 4.217, *p* < 0.001; permutation test, effect size = 0.47, *p* = 0.027, Fig. 4d), suggesting the translation of response pattern to the same outcomes from initial learning to RE after the reversal. These results suggest that lOFC activity represented outcomes of value-guided responses immediately after a reversal which persisted over RE. By contrast, the S1 response pattern represented both the sensory stimulus and the outcome value only after RE. An analogous stimulus-selective RSA analysis for S1, with the onset placed at the time of stimulus presentation, revealed no evidence for different representations of the two alternative stimuli. This suggests that both tactile stimuli shared the same S1 representation due to common sensory features, such as the same stimulation intensity and the same number of stimulating pins (Supplementary Fig. 8).

To identify the distinct topography of stimulus- or outcome-selective response pattern in bilateral S1 and lOFC, we used an RSA searchlight to sweep through the activity in the entire S1_3b and lOFC mask (see 'Methods'). For S1_3b, we specifically asked whether the representation of stimulus- and outcome-selectivity during RE involves ipsi- or contralateral S1. Indeed, we found that, while contralateral S1 selectively represented the stimulus (*p* < 0.005, uncorrected for multiple comparisons, Fig. 5a), the response pattern in ipsilateral S1 selectively represented the outcomes during RE (LE → → RE, *x* = 30, *y* = −36, *z* = 58, $t_{(31)}$ = 4.12, $p_{\text{FWE-SVC}}$ = 0.010, Fig. 5a). This result suggests a disassociated function of bilateral S1 during tactile learning: contralateral S1 is important for stimulus detection as expected, while ipsilateral S1 is, rather unexpectedly, critical for the learning of the stimulus-response association. Furthermore, we identified that the response patterns in right lOFC selectively represented outcomes after the reversal (LE → RN, *x* = 38, *y* = 36, *z* = −10, $t_{(31)}$ = 4.94, $p_{\text{FWE-SVC}}$ = 0.012, Fig. 5b) and bilateral lOFC during RE (LE → → RE, *x* = 50, *y* = 34, *z* = −16, $t_{(31)}$ = 7.21, $p_{\text{FWE-SVC}}$ < 0.001, Fig. 5b).

### The outcome-selectivity in S1 is related to lOFC activity

Combining the univariate and multivariate fMRI data analyses, we demonstrated that responses in the lOFC encoded the prediction error and represented outcome values immediately after a reversal, while ipsilateral S1 exhibited outcome selectivity after RE. Computationally, the function of the prediction error is to update or 'teach' associations between the sensory stimulus and future outcomes, which could be represented by the outcome selective ipsilateral S1. Given that the OFC sends neuroanatomical projections to S1, we next tested whether lOFC responses have influenced the outcome-selectivity of ipsilateral S1 activity. To this end, we performed a connectivity analysis: a psycho-physiology interaction (PPI). This analysis is based on the reasoning that if the outcome-selective S1 activity is dependent on a top-down 'teaching' signal generated in the lOFC, the lOFC, identified in the RSA searchlight analysis, must present enhanced connectivity with the ipsilateral S1 while encoding the outcome-value during RE.

We performed two PPI analyses to test task-related connectivity after the reversal (RN) and during RE by using two different seed regions. The first PPI used the outcome-selective lOFC subregion derived from the RSA searchlight analysis immediately after the reversal (LE → RN, peak MNI coordinates *x*/*y*/*z* = 38/36/−10, Fig. 5b) as

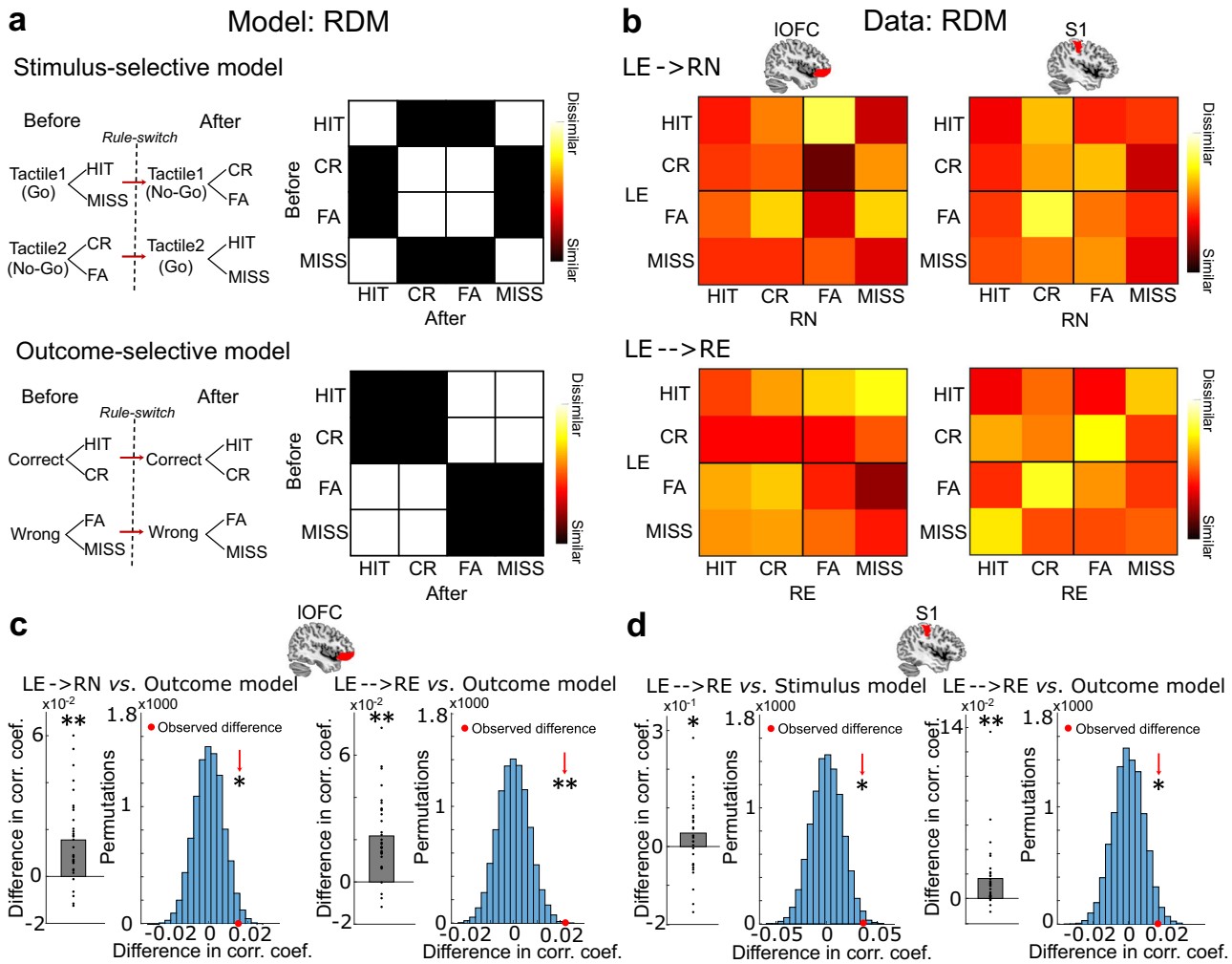

**Fig. 4 | The stimulus- and outcome-selectivity of response patterns in S1 and lOFC, respectively. a** The schematic and representational dissimilarity matrix (RDM) of the stimulus-selective and outcome-selective model. Black elements indicate similarity and white elements indicate dissimilarity between the response pattern before and after reversal. **b** Average RDMs of response pattern in S1 and OFC across all participants for the immediate effect of the reversal (LE → RN) and the stable adaptation after re-learning (LE → → RE), respectively. **c** The response patterns in lOFC significantly represented the outcomes during both the immediate effect of the reversal (LE → RN, signed rank test, $Z_{(31)} = 3.974$, $p = 3.54 \times 10^{-5}$; permutation test, effect size = 0.52, $p = 0.021$) and the stable adaptation after re-learning (LE → → RE, signed-rank test, $Z_{(31)} = 4.628$, $p = 1.85 \times 10^{-6}$; permutation test,

effect size = 0.87, $p = 0.0003$). **d** The response patterns in S1 significantly represented both, the stimulus (signed-rank test, $Z_{(31)} = 1.861$, $p = 0.031$; permutation test, effect size = 0.59, $p = 0.011$) and outcome (signed-rank test, $Z_{(31)} = 4.217$, $p = 1.24 \times 10^{-5}$; permutation test, effect size = 0.47, $p = 0.027$) during the stable adaptation after re-learning (LE → → RE). Signed-rank and permutation tests were applied one-sidedly. One asterisk indicates $p < 0.05$, and two asterisks indicate $p < 0.001$. Source data are provided as a Source Data file. The brain illustrations above figures indicate the lOFC and S1_3b mask superimposed on the sagittal slice of a standard T1-weighted image from the Colin27 brain template implemented in MRIcron. Chris Rorden's MRIcron, all rights reserved.

the seed region. The second PPI used the outcome-selective lOFC subregions derived from the RSA searchlight analysis during re-learning (LE → → RE, peak MNI coordinates $x/y/z = 50/34/-16$, Fig. 5b) as the seed region. We found evidence for a significantly strengthened connectivity immediately after a reversal (RN) between the outcome-selective lOFC subregion and ipsilateral S1 ($x = 20$, $y = -34$, $z = 64$, $t_{(31)} = 4.58$, $p_{FWE-SVC} = 0.013$, Fig. 6a). This S1 subregion largely overlapped with the outcome-selective S1 subregion derived from the RSA searchlight analysis during re-learning (Fig. 5a). In the second PPI analysis, we found no significant changes in the connectivity between the outcome-selective lOFC subregion during re-learning and the S1 area ($p > 0.05$). These findings support the notion that the outcome-selective lOFC conveys a prediction error-related 'teaching' signal immediately after the reversal, which drives the functional configuration of outcome-selectivity in ipsilateral S1 to support behavioral adaptation during RN (Fig. 6b).

## The specificity of lOFC 'teaching' signals to S1

Considering that there were other frontal areas encoding the outcome PE, like MFG (Fig. 2), we tested whether the top-down 'teaching' signals to S1 may have alternatively originated from MFG. To this end, we extracted activity from the MFG mask and performed the same RSA and PPI analyses as for lOFC. We specifically tested whether MFG exhibits analogous neural representations and connectivity patterns with S1 as lOFC. In the RSA analyses, we found that the response patterns in MFG did not significantly represent the stimulus or the outcome after reversals (LE → RN, Supplementary Fig. 9). However, during re-learning, the response pattern in MFG was selective for the outcomes (LE → → RE, signed-rank test, $Z_{(31)} = 4.31$, $p < 0.001$, permutation test, effect size = 0.44, $p = 0.03$), but not for the stimulus (Supplementary Fig. 8). In the PPI analysis, we did not find evidence for a significantly strengthened connectivity between the MFG and S1 immediately after a reversal (RN). Together, these findings provide

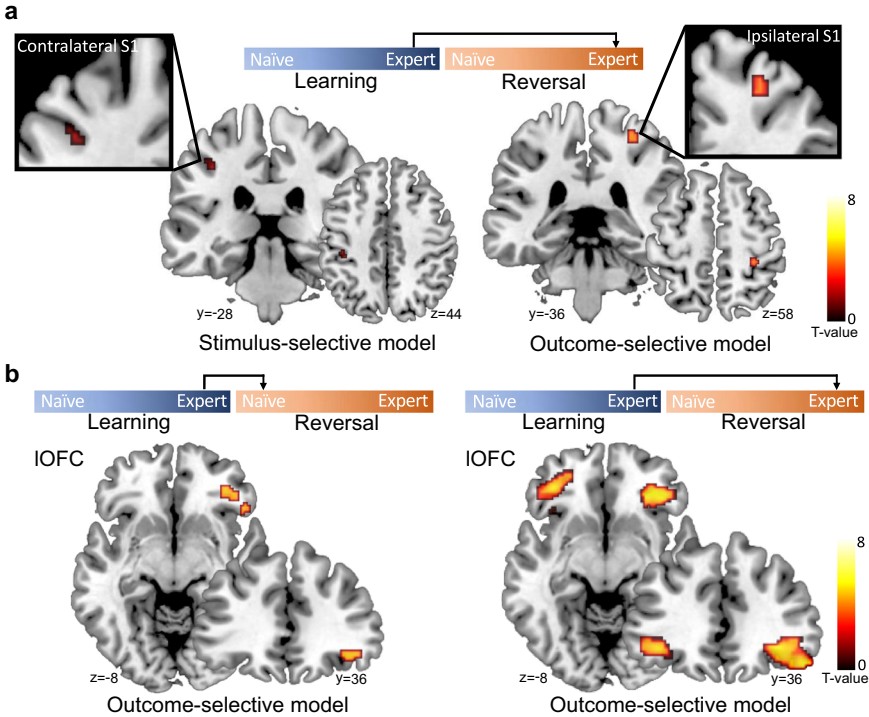

**Fig. 5 | Brain regions within lOFC and S1 masks representing the stimulus- and outcome-selectivity. a** The searchlight revealed that the contralateral S1 represented the tactile stimulus during re-learning (LE → RE, one-sided *t*-test, *p* < 0.005, uncorrected, for display purposes only), while ipsilateral S1 selectively represented the outcomes during re-learning (LE → RE, one-sided *t*-test, $t_{(31)} = 4.12$, *p* = 0.010, family-wise error (FWE) peak-level corrected using small-volume correction (SVC) for multiple comparisons). **b** The searchlight revealed that the right lOFC selectively represented outcomes immediately after the reversal (LE → RN, one-sided *t*-test, $t_{(31)} = 4.94$, *p* = 0.012, *p*eak-level FWE corrected using SVC for multiple comparisons), while bilateral lOFC during re-learning (LE → RE, one-sided *t*-test $t_{(31)} = 7.21$, *p* < 0.001, *p*eak-level FWE corrected using SVC for multiple comparisons). The results were superimposed on coronal and axial slices of a standard T1-weighted image from the Colin27 brain template implemented in MRIcron. Chris Rorden's MRIcron, all rights reserved. Red-yellow coding indicates the *t*-scores of activation intensities.

evidence that the top-down feedback to S1 is specifically related to the lOFC rather than MFG.

## Discussion

Humans and mouse brains can associate sensory stimuli with predicted outcomes by weighing accumulated past and current evidence and flexibly reconfigure the responsiveness to changing environmental demands. Studies investigating such processes in animal models and humans have primarily used reversal learning tasks, where the associations between stimuli and predicted outcomes are initially learned over a series of trials and then reversed. The ability to adapt behavior after the rule reversal is a direct measure of behavioral flexibility[28,29]. By employing a reversal-learning task paradigm, our study elucidates a comparable computational framework underlying adaptive behavior in mice and humans. Our findings show that at the time of outcome presentation, the human lOFC plays a crucial role in encoding deviations from the expected outcome value after a rule reversal, which is essential to achieving behavioral flexibility. By contrast, S1 exhibits a functional dissociation, with contralateral S1 being important for sensory detection and discrimination, whereas ipsilateral S1 represents the outcome value after re-learning. Such functional specialization is generally attributed to higher-order cortices but not the primary sensory cortex. Critically, the prediction error-related lOFC conveys a teaching signal to implement this higher-order functionality into ipsilateral S1 during the re-learning phase (Fig. 6b), perhaps mediated through specific neuronal ensembles[19]. In this context, the PPI analyses we applied are correlative and do not allow us to infer the directionality of information exchange[30]. However, according to the RSA and PPI results shown in Fig. 5 and Fig. 6, we found that outcome-related lOFC activity immediately after the rule switch (i.e.,

reversal) closely related to the outcome-selective ipsilateral S1 signal, that, however, occurred later in the task, after RE. In light of this time order, it is more likely that the outcome-selectivity in lOFC is responsible for shaping outcome-selectivity in ipsilateral S1 than the other way around (Supplementary Fig. 10). Together, our findings extend observations in mice, suggesting that lOFC is specifically involved in assigning credit for unexpected outcomes to specific stimulus–response associations through signaling the outcome values to the sensory cortex, which concurrently results in behavioral adaptation[19].

The contribution of lOFC to flexible decision-making has long been investigated[31,32]. Studies with lOFC lesions in monkeys and rodents have commonly found that orbitofrontal damage does not impair the initial learning of stimulus-response associations but instead impairs the learning of stimulus-outcome reversals[19,29]. Similarly, in humans, using a simple deterministic reversal learning task, damage to lOFC was particularly associated with decreased adaptations during the reversal phase of the task[33]. However, in a more challenging and dynamic probabilistic environment, lOFC damage disrupted both initial and reversal learning[34]. In our analysis of human lOFC responses across all behavioral phases, we revealed its prominent engagement in both naïve periods (LN and RN). However, in mice, pronounced lOFC engagement was observed in RN but not in LN[19]. This may be due to the probabilistic nature of the reversal learning task in humans, whereas a deterministic paradigm was applied to mice. In probabilistic contexts, the accurate choice of actions requires the integration of previous feedback history, titrated to the particular reinforcement structure of the task[34]. The lOFC has been suggested to play a general role in using such feedback about the outcome history across trials to adjust behavior. In this regard, the reversal of the

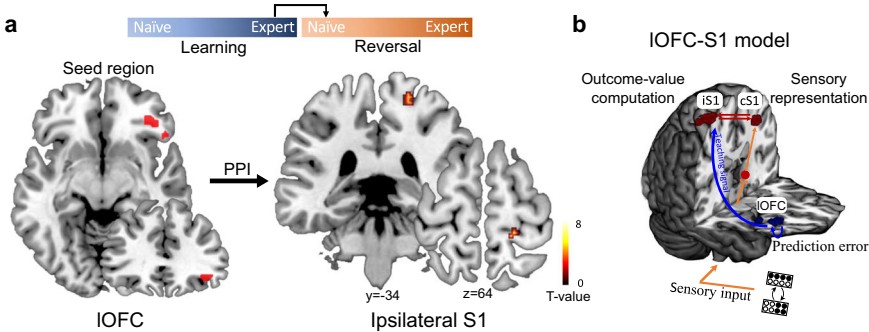

**Fig. 6 | Connectivity between outcome-selective lateral OFC and ipsilateral S1.**
**a** Psychological-physiological interaction (PPI) shows significantly strengthened connectivity between lateral OFC (seed region, MNI coordinates x/y/z = 38/36/−10]) and ipsilateral S1 (peak MNI coordinates x/y/z = 20/−34/64]) immediately after a reversal (RN > LE, one-sided *t*-test, $t_{(31)}$ = 4.58, *p* = 0.013, peak-level FWE corrected using SVC for multiple comparisons). The results were superimposed on coronal and axial slices of a standard T1-weighted image from the Colin27 brain template implemented in MRIcron. Chris Rorden's MRIcron, all rights reserved. Color coding indexes the *t*-scores in each voxel. **b** Schematic showing the dynamic interaction between lateral OFC and S1. While the stimulus-selective contralateral S1 (cS1) receives the sensory input from the right index finger (follow orange arrows), the lateral OFC (blue blob) sends a prediction-error related "teaching signal" to assign outcome values to sensory inputs in ipsilateral S1 (iS1). The 3D brain was created using MRIcron. Chris Rorden's MRIcron, all rights reserved.

stimulus–outcome association is simply one instance of a general requirement for behavioral adjustment based on expectancy violation[35].

The topographic assignment of the 'lOFC' region we identified in the context of outcome-selectivity is approximate and should be further confirmed. In the literature, especially in macaque monkeys, who present a comparable prefrontal architecture as humans[36,37], diverse anatomical nomenclatures were used. Rudebeck et al., for instance, defined a broad orbital-lateral prefrontal area as ventrolateral prefrontal cortex[38,39], which was involved in credit assignment, and which encompasses the monkey equivalent of the outcome-selective region we identified between the foci of OFC and lateral prefrontal cortex. Folloni et al. also identified credit assignment-related functions in a comparable albeit smaller ventrolateral prefrontal region[40], which they referred to as 47/12o according to Brodmann's cytoarchitectural brain atlas, and which substantially overlaps with the regions we found in the context of outcome-selectivity (see Fig. 2 and Fig. 5). In our study, we referred to this outcome-selective prefrontal region as lOFC, but the corresponding neural activity we found in this area may have slightly extended into adjacent areas, such as the dorsolateral prefrontal cortex. We cannot rule out that these areas may have also contributed to generating decision outcome signals.

During stimulus presentation, S1 presented rather weak stimulus-selective responses, but during the outcome phase, we found strong stimulus- and outcome-selective S1 responses. These representations in S1 are important for linking the outcome value to the corresponding sensory stimulus, which is in line with previous evidence suggesting that outcome feedback is associated with activity in sensory areas involved in stimulus processing during outcome presentation, even in the absence of concurrent sensory stimulation[13,14,16,41]. One way the brain might perform this operation is to encode and transmit a 'teaching' signal, based on both positive and negative outcomes, to sensory regions involved in stimulus processing[10,11]. PFC maintains neuroanatomical connections with sensory cortices to support value-guided decision-making[3,42]. In rodents, the cingulate cortex, often considered as part of the dorsal medial PFC, directly influences sensory processing in the primary visual cortex (V1) through long-range projections[43]. Similarly, in primates, lesions of lateral PFC reduce attentional modulation, suggesting that the PFC is necessary for attention-related control of visual cortical responses[44]. Signals from OFC have also been proposed to be necessary for the detection of prediction errors, to update or 'teach' the associative representations in sensory cortices when stimulus associations change[9]. The involvement of OFC in the modulation of sensory responses also draws support from rodent studies. Ventrolateral OFC neurons, for instance, that maintain projections to V1 have been shown to mediate the outcome-expectancy modulation of V1 responses to the reward-irrelevant stimulus—a process that is required to drive visual associative learning[17]. Similarly, a direct connection from ventrolateral OFC to the primary auditory cortex (A1) is capable of shaping A1 receptive fields and thereby enhancing sound processing[18]. These studies show that OFC projections to the sensory cortex are important for understanding how sensory representations are dynamically adjusted to reflect changing behavioral relevance of incoming stimuli. Using the reversal learning task in rodents, Banerjee et al. revealed that the encoding of outcome value by the lOFC is essential to the functional remapping of S1 neurons in support of flexible decision-making[19]. To our knowledge, this has rarely been tested in humans. One previous study used patients with lesions in the OFC and suggested that the OFC exerts top-down attentional control to modulate auditory sensory processing[45]. Here, we directly tested the notion that the lateral part of OFC can especially influence the sensory cortex in humans, which renders the lOFC an essential player in assigning outcome values to sensory stimuli and facilitating the encoding of new associations in sensory areas to adapt associated behavior[9]. Except for lOFC, other prefrontal areas, such as the anterior cingulate cortex, ventromedial prefrontal cortex, or integrative brain areas like the posterior parietal cortex, may exert comparable or complementary interaction with S1 in the context of reversal learning, which can be further explored in future studies. Our additional analyses regarding the specificity of lOFC involvement provided insight into this question. Specifically, the response pattern in the MFG, another frontal area encoding the PE (Fig. 2), represented outcome-selectivity only during re-learning and, importantly, was not related to S1 signals, unlike the lOFC (Supplementary Fig. 8). Nevertheless, we hypothesize that lOFC feedback may directly engage sensory areas, but comparable interactions can also involve other integrative areas[12]. The involvement of these areas may, however, be species-specific and task contingency dependent[46].

Interestingly, we provide evidence for distinct functional engagement of bilateral S1 in humans: contralateral S1 is primarily implicated in sensory processing, while ipsilateral S1 is implicated in post-sensory, higher-level cognitive processing. Specifically, ipsilateral S1 receives the 'teaching' signal from lOFC to represent the outcome value for the learned stimulus-response association. Ipsilateral S1 activity in response to unilateral tactile inputs has been shown in both humans[47,48] and monkeys[49,50]. Its role in sensory-cognitive processing, however, remains poorly understood. Most studies interpreted this activity as the transcallosal projection of sensory processing from

contralateral S1[47,51]. But a transcallosal route directly affecting area 3b is highly unlikely because area 3b is practically free of transcallosal connections[52] even though ipsilateral area 3b can obtain tactile input through transcallosal connections, most likely through area 2, which has the densest transcallosal connections among all S1 areas[52]. However, S1 also receives projections from other higher-order cortical regions, such as the PFC, but the top-down influences on ipsilateral S1, especially ipsilateral area 3b, and the functional meaning of these modulations are largely unknown. Our present study provides insights into the functional relevance of the ipsilateral S1, which implements computations through the dynamic interaction with lOFC to support flexible decision-making and adaptive behavior.

Taken together, combining human fMRI with a comparable analytic framework as recently applied to neuronal population recordings in mice[19], we revealed dynamic interactions of lOFC with the sensory cortex for the implementation of computations mandatory for flexible decision-making. Given that a lack of behavioral flexibility is a hallmark of many mental illnesses, such as schizophrenia, autism, and obsessive-compulsive disorder[2], our findings have implications for targeting orbitofrontal circuits with non-invasive or invasive neuromodulation to potentially provide a viable strategy for augmenting cognitive and behavioral abilities in brain disorders in the future.

## Methods

### Participants
The required sample size was estimated using the free-source software G*Power (version 3.1.9.2) with a two-tailed t-test between two dependent samples. Based on our previous study[8], we expected a large effect size of 0.7. The error probability was set to 0.05. The predicted sample size was 29. Considering possible exclusions of participants, we recruited 40 participants (22 females, mean age ± SD: 24.5 ± 3.3 years). Participants self-reported their sex. All participants were right-handed and had normal or corrected-to-normal vision. Participants with a history of psychiatric or neurological disorders and those taking regular medication were excluded. The study was approved by the local ethics committee of the Ruhr-University Bochum. All participants gave written informed consent prior to participation.

Two participants were excluded because of technical problems with the fMRI scans, and another two were due to failed training. Thirty-six participants successfully performed the task during fMRI scanning. Data from four participants were excluded from further analyses due to failed learning of the task inside the MRI scanner. Therefore, the data from the remaining 32 participants were further analyzed (16 females, mean age ± SD: 24.5 ± 3.5 years).

### Tactile stimuli
The tactile stimuli were generated and delivered using an MRI-compatible Braille device (Metec, Stuttgart, Germany). The device consisted of eight plastic pins, aligned in two series of four pins (pin diameter 1.2 mm, rounded top, inter-pin spacing 2.45 mm) (Fig. 1a, left upper corner). We created eight alternative tactile stimulation patterns (Fig. 1b), which always consisted of four raised and four lowered pins. Stimuli were applied to the index fingertip of the right (dominant) hand. The Braille device was controlled using the Presentation software (version 20.1, Neurobehavioral Systems, Berkeley, CA, USA) through the Metec Virtual Braille Device by TCP-IP commands. To ensure that all tactile stimulation patterns were correctly perceived, participants performed a tactile detection test prior to the task training and fMRI scan. During the test, participants had to report which pattern they received until they perceived and distinguished all tactile stimulation patterns 100% correctly.

### Experimental design
We employed a probabilistic reversal learning Go/NoGo task. The task was organized in blocks of 45 trials and consisted of 3 runs, each including four blocks. In each block, two tactile patterns were randomly selected from the eight alternative patterns (one 'Go' pattern and one 'NoGo' pattern). In each trial, participants were instructed to maintain central fixation. Participants received one out of the two tactile stimulation patterns for 500 ms on the index fingertip of the right (dominant) hand. A red fixation cross was simultaneously presented on a screen via MRI-compatible LCD goggles (Visuastim Digital, Resonance Technology Inc., Northridge, CA, USA). Following the tactile cue, the red fixation cross turned green, instructing the participants to press the button (LumiTouch keypads, Photon Control Inc., Burnaby, BC, Canada) with the index finger of the left hand ('Go') or refrain from pressing the button ('NoGo'). Participants were instructed to press the button within 1000 ms if action was needed. After the interval of 500-1500 ms, the outcomes were presented for 500 ms to indicate whether the choice was rewarded or non-rewarded. Trials were presented with randomized intertrial intervals ranging between 1500 and 3000 ms in 100 ms steps. A novel pair of tactile patterns was used on each new block, which was presented to the participants at the beginning of each block.

In each block, 70% of trials with one tactile pattern were assigned to 'Go', and 70% of trials with the alternative tactile pattern were assigned to 'NoGo'. By trial and error, participants had to learn which of the two available options ('Go' and 'NoGo' response) had the higher reward probability for each of the two tactile patterns. Importantly, in each individual block, the association between tactile stimuli and responses was switched at a random trial (reversal) within a window from trials 20 to 25. From that point on, participants had to reverse their choice behavior to maximize reward. Participants were told in advance that the association between tactile stimuli and response is probabilistic and that there would be a rule switch in each block, but they were not informed about the levels of probability or when the switch occurs.

To enhance motivation throughout the experiment, we offered a monetary reward of 1€ added to the general reimbursement (5€/run) for a 5% increase in behavioral performance in each fMRI run. After each run, participants were given visual feedback (10 s) about their proportion of correct responses and how much money they made during the preceding run.

Before the fMRI experiment, each subject completed a short and easy practice block with 90% probability instead of 70% to ensure they were able to follow the instructions. The fMRI experiment consisted of 540 trials overall, which we split into three runs, each lasting about 16 min, resulting in a total scanning time of ~50 min.

### Modeling of human behavior
We applied four different computational models to the behavioral data to probe how participants made choices based on previous decision outcomes.

**Model 1: Random responding.** In the first model, we assumed that participants did not engage in the task at all and simply pressed buttons randomly. This random behavior may occur, especially when participants get lost or when they do not have external incentives and motivation to perform well. Modeling such behavior can be used to assess the chance level, which can be compared to more strategic and practical models. To this end, we assumed that participants randomly chose between the two options (stimulus1-Go/stimulus2-NoGo or stimulus1-NoGo/stimulus2-Go in our case), probably with some overall bias for one option over the other. This bias is captured with a free parameter $b$ (which is between 0 and 1), such that the probability of choosing the two options is

$$p_t^1 = b \text{ and } p_t^2 = 1 - b. \tag{1}$$

**Model 2: Noisy win-stay-lose-switch.** The win-stay-lose-switch model is one of the simplest models that updates the decision according to feedback. This model, as the name implies, repeats the choice if the previous action is rewarded and switches if it is unrewarded. In the noisy version of this model, the win-stay-lose-switch rule is applied with the probability of $1-\varepsilon$, and the free parameter $\varepsilon$ is randomly chosen. In the two-choice case, the probability of choosing option k is

$$p_t^k = \begin{cases} 1-\varepsilon/2 \text{ if } (c_{t-1}=k \text{ and } r_{t-1}=1) \text{ or } (c_{t-1}\neq k \text{ and } r_{t-1}=0) \\ \varepsilon/2 \text{ if } (c_{t-1}\neq k \text{ and } r_{t-1}=1) \text{ or } (c_{t-1}=k \text{ and } r_{t-1}=0) \end{cases} \quad (2)$$

where $c_t=1,2$ is the choice in trial $t$, and $r_{t-1}=0,1$ the outcome (wrong or correct) in the previous trial.

**Model 3: Rescorla-Wagner.** In this model, participants are assumed to learn the expected value of each choice based on the history of previous outcomes and then use the updated values of choices to make a decision. The central idea behind the RW learning model is that it quantifies the evaluation of a choice option updated by the difference between the actual outcome and the expected outcome[53]:

$$V_t^k = V_{t-1}^k + \alpha\left(r_{t-1}-V_{t-1}^k\right). \quad (3)$$

Where $\alpha$ is the learning rate, which takes a value between 0 and 1. This captures the extent to which the prediction error, the difference between the actual outcome ($r_{t-1}$) and the expected outcome ($V_{t-1}^k$), updates the value of an option $V_t^k$.

A simple model of how to choose the action is to assume that participants use the updated values of an option $V_t^k$ to guide their decisions. This implies that the most valuable option is chosen most frequently, but occasionally 'mistakes' (or exploring) occur due to choosing a low-value option. The 'softmax' choice rule describes these properties, which chooses option k with the probability

$$p_t^k = \frac{\exp\left(\beta V_t^k\right)}{\sum_{i=1}^K \exp\left(\beta V_t^i\right)}. \quad (4)$$

Where $\beta$ is the 'inverse temperature' parameter that controls the level of stochasticity in the choice, ranging from $\beta=0$ for completely random choices and $\beta=\infty$ for deterministic choices in favor of the highest value option. Combining the learning (Eq. 3) and decision rules (Eq. 4) gives a simple model of decision-making in this task with two free parameters: the learning rate, $\alpha$, and the inverse temperature, $\beta$.

**Model 4: Hierarchical Gaussian Filter.** The Hierarchical Gaussian Filter (HGF) consists of a perceptual and a response model, which describes a framework where an agent receives a sequence of inputs (stimuli) and generates behavioral responses based on Bayesian inference. The perceptual model we used in the present study is the two-level version of the HGF (v7.0, https://www.tnu.ethz.ch/de/software/tapas.html), where we eliminated the third level, i.e., the log-volatility of the environment, from the hierarchy by fixing both, the value of log-volatility $\vartheta$ and couple strength $k$ between second and third levels, to zero. The two-level version of HGF model assumes a low or stable volatility over the time course of the experiment, which is in line with our experimental setting where the participants were informed about only one reversal in each block.

The first level of the perceptual model represents a sequence of environmental states, $x_1^{(t)}$. In our study, it was represented by a binary input, with $x_1^{(t)}=1$ for the stimulus1 → 'Go'/stimulus2 → 'NoGo' and $x_1^{(t)}=0$ for the stimulus1 → 'NoGo'/stimulus2 → 'Go'. The second level $x_2^{(t)}$ represents the beliefs about the stimulus-response association, i.e., the conditional probability of receiving a reward when performing a

Go or NoGo response, given the presence of stimulus1. The model assumes that the variance of the environmental hidden states depends on the state at the next higher level changing as a Gaussian random walk[54] as follows:

$$p(x_1|x_2) = s(x)^{x_1}(1-s(x_2))^{1-x_1} = \text{Bernoulli}\left(x_1;s(x_2)\right). \quad (5)$$

$$p\left(x_2^{(t)}|x_2^{(t-1)},x_3^{(t)}\right) = N\left(x_2^{(t)};x_2^{(t-1)},\exp(\omega)\right). \quad (6)$$

Where $t$ is a trial index, and $s$ is a sigmoid function as follows:

$$s(x) = \frac{1}{1+\exp(-x)}. \quad (7)$$

At the second level (Eq. 6), the step size between consecutive time steps depends on $\omega$, which is a free parameter of the perceptual model in HGF.

Under a variational approximation to ideal hierarchical Bayesian learning according to the above equations, at any level $i$ of the hierarchy, the update of the belief on trial $t$ (i.e., posterior mean $u_2^{(t)}$ of the state) at the second level is proportional to the outcome prediction error $\delta_1^{(t)}$ weighted by the precision of predictions $\varphi_2^{(t)}$:

$$u_2^{(t)} = u_2^{(t-1)} + \varphi_2^{(t)}\delta_1^{(t)} \quad (8)$$

The precision weight $\varphi_2^{(t)}$ is updated with every trial and can be regarded as equivalent to a dynamic learning rate in reward learning models, as follows:

$$\varphi_2^{(t)} = \frac{1}{1/\hat{\varphi}_2^{(t)}+\hat{\varphi}_1^{(t)}} \quad (9)$$

$$\hat{\varphi}_1^{(t)} = \hat{u}_1^{(t-1)}\left(1-\hat{u}_1^{(t-1)}\right) \quad (10)$$

$$\hat{\varphi}_2^{(t)} = \varphi_2^{(t-1)} + e^{\omega} \quad (11)$$

The outcome prediction error $\delta_1^{(t)}$, which drives learning at the second level of our HGF model, is defined as the difference between the actual outcome and its estimated probability before the outcome:

$$\delta_1^{(t)} = u_1^{(t)} - \hat{u}_1^{(t)} \quad (12)$$

$$\hat{u}_1^{(t)} = s\left(u_2^{(t-1)}\right) \quad (13)$$

Therefore, while the updating in HGF learning is structurally similar to that in the RW model, the HGF model differs fundamentally from the RW in that prediction errors are weighted by time-dependent precision weights instead of a constant learning rate[55].

Notably, the sign of the outcome prediction error in contingency space depends on the arbitrarily chosen coding of a binary input (in our case, the assignment of $x_1^{(t)}=1$ for the stimulus1 → 'Go'/stimulus2 → 'NoGo' and $x_1^{(t)}=0$ for the stimulus1 → 'NoGo'/stimulus2 → 'Go'). In this study, we used the unsigned outcome prediction error (i.e., absolute value) that corresponds to Bayesian surprise[56,57], and which is equivalent to the prediction error we investigated in our previous rodent study[19].

The observation (response) model describes how the states or values of the perceptual model map onto responses. In the HGF model, we used the unit-square sigmoid to simulate the responses, which maps the prior belief $\hat{u}_1^{(t)}$ onto the probabilities $p_t^{k=1}$ and $p_t^{k=0}$ that the

agent will choose response 1 or 0, respectively:

$$p_t^k = \left( \frac{\hat{u}_1^\zeta}{\hat{u}_1^\zeta + (1+\hat{u}_1)^\zeta} \right)^k \times \left( \frac{(1+\hat{u}_1)^\zeta}{\hat{u}_1^\zeta + (1-\hat{u}_1)^\zeta} \right)^{1-k}. \qquad (14)$$

Like the softmax decision rule uses the free parameter $\beta$ (Eq. 4) to control the level of stochasticity in the choice, our decision model uses a constant free parameter $\zeta$ that captures how deterministically the response is associated with the prior belief $\hat{u}_1^{(t)}$. The higher $\zeta$, the more likely the agent chooses the option that is more according to its current belief. Therefore, there are two free parameters in the HGF: $\omega$ in the perceptual model and $\zeta$ in the response model.

## Model simulation, fitting, and comparison

To test how participants switched their decision strategy based on the previous decision outcome, we first simulated the responses of four models given a set of particular parameters which were independent of the actual behavioral responses and compared them with the participants' actual behavioral responses. To this end, we simulated responses across a range of parameter settings to determine how the model-independent measures change with different free parameters. Based on these model simulations, the parameters for the response module in the four models (Random responding: b; WSLS: $\varepsilon$; RW: $\beta$; HGF: $\zeta$) were determined. For the free parameters of the additional perceptual module in the RW ($\alpha$) and HGF ($\omega$) model, we utilized the 'Bayes optimal' values, which are the optimizations that produce the least cumulative Shannon surprise for a given input sequence $\mu$ based on a free energy minimization approach. Therefore, an agent utilizing these parameter settings would experience the least possible surprise when exposed to the given inputs in the given perceptual model. Please note that these 'Bayes optimal' values are independent of the participant's actual behavioral responses. Using the four different models with these particular parameters, we simulated the responses and tested one measure that captures fundamental aspects of the flexible decision process based on prior experience: the probability of repeating a decision, $p$(staying). We repeated the simulation process 1000 times and plotted $p$(staying) as a function of the outcomes (correct or wrong) of the previous trial for each of the four models.

Next, we fitted two alternative models (RW and HGF) to the participants' actual behavioral responses to separately estimate the maximum a posteriori estimates of the free parameters. The fitting procedure started at a random initialization of the optimal value and proceeded iteratively as the value converged to the solution based on a computationally efficient quasi-Newton minimization algorithm (i.e., the Broyden–Fletcher–Goldfarb–Shanno or BFGS algorithm) implemented in the HGF toolbox. By fitting the model to the participants' actual behavioral responses, the measures of model goodness (i.e., LME) were also calculated. The LME is calculated as the negative variational free energy under the Laplace assumption. LMEs can be used to calculate Bayes factors by exponentiating the difference in LME between two models applied to the same dataset. To identify the model that best explained participants' behavior (RW or HGF), we applied Bayesian Model Selection (BMS). BMS is a standard approach in machine learning and computational neuroimaging[58] that compares different models to infer how neurophysiological or behavioral responses were generated. BMS assesses the relative plausibility of competing models based on their log evidence, which represents the negative surprise about the data given the model and quantifies the trade-off between accuracy (fit) and model complexity. We used random effects BMS to account for potential interindividual variability in our dataset[59] and to quantify the posterior probabilities of the two competing models (RW and HGF). Using the subject-specific parameters optimized based on participants' actual behavioral responses,

the individual PE trajectories were finally assessed using the winning HGF model (see 'Results' section).

## fMRI data acquisition

We collected the fMRI data on a Philips Achieva 3.0 T X-series scanner using a 32-channel head coil. Functional scans were collected using a multi-band echo-planar imaging (EPI) sequence with a multi-band acceleration factor of 2. Thirty-eight transaxial slices parallel to the anterior-posterior commissure (AC-PC) covering the whole brain were acquired with a voxel size of $2 \times 2 \times 3$ mm$^3$, TR = 2200 ms, TE = 24 ms, flip angle = 90, the field of view 224 mm, and no interslice gap. For each participant, high-resolution T1-weighted structural images were acquired, with 176 transversally oriented slices covering the whole brain, to correct for geometric distortions and perform co-registration with the EPIs (isotropic T1 TFE sequence: voxel size: $1 \times 1 \times 1$ mm$^3$, the field of view $240 \times 176$ mm$^2$).

## fMRI data preprocessing and GLMs

For each run, we acquired a total of 453 EPI volumes. To allow for T1-equilibration, five dummy scans preceded data acquisition in each run, which were removed before further processing. Each participant's EPI volumes were preprocessed and analyzed with the Statistical Parametric Mapping software SPM12 (Wellcome Department of Imaging Neuroscience, University College London, UK; http://www.fil.ion.ucl.ac.uk/spm) implemented in MATLAB R2017b (MathWorks Inc.). For preprocessing, EPI images were first realigned to the first volume and corrected for distortion using field maps. Then, the T1w image was normalized to the Montreal Neurological Institute (MNI) reference space using the unified segmentation approach. Subsequently, the resulting transformation was applied to the individual EPI volumes to transform the images into standard MNI space and resample them into $2 \times 2 \times 2$ mm$^3$ voxel space. Spatial smoothing with a 6-mm FWHM Gaussian kernel was applied to the fMRI images only for the univariate general linear model (GLM) and psychophysiological interaction analysis (see below) but not for RSA analyses. Data were high pass filtered at 1/128 Hz to remove low-frequency signal drifts. For each participant, the preprocessed fMRI data were analyzed in an event-related manner in three GLMs, one designed for univariate analyses, a second designed for multivariate analyses (RSA), and a third designed for assessing functional connectivity using psychophysiological interaction (PPI). In all GLMs, six head-motion parameters as estimated during the realignment procedure, were defined as regressors of no interest to account for motion-related artifacts during the task.

The first GLM used to analyze the univariate BOLD effect included four regressors of interest per block, which accounted for trials in the four different phases of the task (LN, LE, RN, RE). This univariate GLM was used to test overall changes in BOLD responses after the reversals (i.e., RN > LE) and during re-learning (i.e., RE > LE). To assess the prediction error-related signals, we included two parametric modulators in the univariate GLM. For each of these four main regressors, the absolute value of trial-by-trial outcome prediction error ($\delta_I^{(t)}$) derived from the HGF model was defined as a parametric modulator. To control the effect of outcomes on the prediction error signals, another parameter that accounted for the outcomes (i.e., reward/no-reward; 1/0) was added. In this model, we switched off orthogonalization to consider the collinearity of the two parameters. The onset was time-locked to the outcome presentation of each trial. Two additional regressors of no interest accounted for the presentation of the stimuli (all trials collapsed to a single regressor, time-locked to the onset of cue presentation) and invalid trials (i.e., late responses). All regressors were then convolved with the canonical hemodynamic response function in an event-related fashion. The design matrix orthogonality in SPM showed only weak between-events correlations between the stimulus regressor and the outcome regressors (the value of the cosine of angle: mean ± SD = 0.22 ± 0.05, Supplementary Fig. 11), which

suggests that our design allowed to separate outcome-related from stimulus-related brain responses.

The second GLM used to assess the representational similarity between different phases of learning using RSA, consisted of the unsmoothed fMRI data separated into 16 regressors of interest per block. These 16 regressors accounted for trials of the four different phases of the task (LN, LE, RN and RE), divided into the different outcomes (HIT, Correct Rejection or CR, False Alarm or FA and MISS). The onset of events for these 16 regressors were time-locked to the onset of the outcome in each trial. The same applied to the first GLM, where two additional regressors of no interest were included (i.e., presentation of the stimuli and invalid trials). All regressors were then convolved with the canonical hemodynamic response function in an event-related fashion.

The third GLM applied to assess functional connectivity using PPI included five regressors of interest, consisting of physiological, psychological, and PPI regressors. The physiological regressor was defined as the fMRI time series extracted from a seed region. Two psychological regressors accounted for trials before and after the reversal (i.e., LE&RN or LE&RE). Two PPI regressors accounted for the interactions between the physiological variable and psychological regressors by extracting and deconvolving the time series from the seed region, multiplying it by the psychological regressor, and then convolving the output with the hemodynamic response function. To account for additional unwanted variance, we also included two regressors representing the presentation of the stimuli and invalid trials.

## Univariate fMRI analysis

Using the first GLM for univariate analysis, the prediction error-related parametric effects were first assessed by applying the trial-by-trial prediction error derived from the HGF model to the four main regressors. Next, two contrasts were assessed to reveal changes in BOLD responses after the reversal of stimulus–response association. First, to measure the BOLD response to the immediate effect of the reversal, the fMRI BOLD signal during Reversal Naïve (RN) trials was contrasted with the fMRI BOLD signal during LE trials. Second, to measure the BOLD response to the adaptation after re-learning, Reversal Expert (RE) trials were contrasted with LE trials. The contrast images (i.e., "RN > LE" and "RE > LE") were next applied to the group-level one-sample $t$-test and thresholded at $p = 0.05$, FWE-corrected. Based on the study in mice[19], we hypothesized that the immediate effect of the reversal ("RN > LE") and the stable adaptation after re-learning ("RE > LE") is related to the lateral OFC and bilateral S1, respectively. Therefore, we performed small volume correction (SVC) by restricting the search volume to lateral OFC and entire S1 regions. To this end, we created lateral OFC and S1 masks, as implemented in the SPM Anatomy Toolbox[25,26]. The GLM results were superimposed on sagittal, coronal, and axial slices of a standard T1-weighted image from the Colin27 brain template implemented in MRIcron (https://www.nitrc.org/projects/mricron).

## Representational similarity analysis

To investigate whether the multi-voxel response pattern in lateral OFC and S1 before the reversal is translated into a representation of the same tactile stimulus (stimulus-selective) or a representation of the same outcome (outcome-selective) after reversal, we performed an RSA. Multi-voxel measures of neural activity are quantitatively related to each other and to computational theory and behavior by comparing RDMs.

## Construction of model RDMs

Based on the predicted correlation distance for trials before and after reversal, two model RDMs were constructed to investigate whether the multi-voxel response pattern in lateral OFC and S1 at the time of outcome presentation is stimulus-selective or outcome-selective. The stimulus-selective model describes how the response pattern to a tactile stimulus before reversal shows higher representational similarity with the trials associated with the outcomes of the same tactile stimulus after the reversal (i.e., $HIT_{learning} = CR_{reversal}$). The outcome-selective model describes how the response pattern to the outcomes before reversal shows higher representational similarity with the trials associated with the same outcomes after the reversal (i.e., $HIT_{learning} = HIT_{reversal}$).

**Construction of ROI RDMs.** Based on the univariate fMRI analysis, we defined two ROIs, lOFC and S1_3b, respectively, as derived from the SPM Anatomy Toolbox. Using the output of $t$-statistic maps from the second GLM, activity patterns were extracted from lOFC and S1_3b masks. The relative similarity between the response patterns, elicited in different trials, was assessed using Pearson correlation and expressed as a correlation coefficient. For each participant, the response patterns from trials before reversal were compared with the response patterns from trials after reversal. Note that unlike a distance or a correlation matrix, this matrix is not symmetric. To assess both the immediate effect of the reversal and the stable adaptation after re-learning, we compared the trials after reversal with the trials during LE twice (immediate effect RDM: RN vs. LE; stable effect RDM: RE vs. LE), resulting in two RDMs for each participant and for each ROI.

**ROI analysis.** The response pattern in S1 and lOFC during initial learning (LE) and after the reversal (RN, RE) were compared using RSA to establish a cross-phase representational dissimilarity matrix (RDM) as described above. We also estimated the mean 'similar' (black elements in model RDMs) versus the mean 'dissimilar' (white elements in model RDMs) for both the immediate effect RDM and the stable effect RDM of each ROI separately. Summary statistics were tested at the group level using two approaches: (1) one-sided Wilcoxon signed-rank test across participants; (2) one-sided permutation test where the null distribution was generated by estimating the group average 10,000 times, after permuting the identity of trials in the RDM on each iteration.

**Searchlight analysis.** We also conducted a searchlight analysis with a radius of 6 mm relative to the center voxel within the entire OFC and S1 ROI using the RSA toolbox. Across $t$-statistic maps, the extracted voxels were correlated using Pearson correlations and expressed as a correlation coefficient. The RDM was then constructed as described for the ROI RDM analysis above. A summary statistic was then generated for each searchlight sphere using the model RDM to estimate stimulus or outcome selectivity. The summary statistic of interest was then mapped back to the central voxel in the searchlight sphere and saved. The sphere was then shifted, and the whole procedure was repeated until complete for the entire ROI mask. This yielded two separate descriptive maps per participant—one for the immediate effect of the reversal (LE → RN) and the other for the stable adaptation after re-learning (LE → → RE). Each participant's correlation maps with the model RDMs were spatially smoothed with a 6-mm FWHM Gaussian kernel and entered into the second-level random-effect analysis performed in SPM12 across the group. The statistical significance at the group level was thresholded at $p < 0.05$ with a voxel-level FWE small-volume correction within the lateral OFC and S1 ROIs.

## Psychophysiological interaction

PPI was used to assess context-related differences in functional connectivity between a given seed region and the rest of the brain. We performed PPI analyses to assess changes in connectivity between trials before reversal (LE) and after the reversal (RN and RE) using the generalized PPI (gPPI) toolbox. Since RSA results revealed that the response pattern of lateral OFC and ipsilateral S1 were outcome-selective, we applied three PPIs, the first using the OFC as the seed to

investigate the immediate effect after the reversal (RN vs. LE) and the second and third using either ipsilateral S1 or OFC as the seed region respectively to investigate the stable period after re-learning (RE vs. LE).

Individual time series of each seed region were extracted from ROIs that were identified with the RSA searchlight analyses of the outcome-selective lateral OFC and ipsilateral S1 within a radius of 12 mm from the group maximum. The first Eigenvariate was then calculated across all voxels surviving $p = 0.05$, uncorrected, within a 6 mm sphere centered on the individual peak voxel. The resulting BOLD time series were adjusted for effects of no interest (e.g., invalid trials and movement parameters) and deconvolved to generate the time series required for constructing first-level GLMs for the PPIs as described in the "fMRI data preprocessing and GLMs" section.

First, we examined the immediate effect of the reversal on lOFC connectivity. To this end, first-level contrast images were created using the PPI regressor of the interaction between the physiological variable and LE trials, as well as the interaction between the physiological variable and RN trials. Next, the contrast images (i.e., RN > LE) were applied to the group-level one-sample $t$-test. We hypothesized that the immediate effect of reversal was related to interactions between the OFC and S1. Therefore, we performed a small volume correction by restricting the search volume to the S1 mask. Second, to test the stable period of re-learning after the reversal, two PPIs were performed using either S1 or OFC as the seed region, respectively. For each ROI, the first-level contrast images were created using the PPI regressor of the interaction between the physiological variable and LE trials, as well as the interaction between the physiological variable and RE trials. The contrast images (i.e., RE > LE) were next applied to the group-level one-sample $t$-test. Small volume correction was used by restricting the search volume to either the OFC or the S1_3b mask with a threshold at an FWE-corrected peak level of $p < 0.05$.

### Reporting summary

Further information on research design is available in the Nature Portfolio Reporting Summary linked to this article.

## Data availability

The behavioral data and processed fMRI data have been deposited at Sciebo and are publicly available as of the date of publication. The raw imaging data is not publicly available due to restrictions related to the individual information that could compromise the privacy of research participants. Source data are provided in this paper.

## Code availability

The custom codes for data analysis have been deposited in a GitHub repository. The version of the code used in this study was also archived in the Zenodo repository[60].

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

## Acknowledgements

This work was supported by the Deutsche Forschungsgemeinschaft (DFG, German Research Foundation): Project number 122679504—SFB 874 'Integration and Representation of Sensory Processes' (to B.A.W. and B.P.), project number PL602/6-1 Prefrontal-thalamic control of cognitive flexibility—from mice to humans (to B.P.), a Wellcome Trust institutional strategic award (RES/0100/7524/220; to A.B.), a Royal Society research grant (RGS\R2\202155; to A.B.), and National Natural Science Foundation of China: Project number 32200867 (to B.A.W.). We acknowldege support by the Open Access Publication Funds of the Ruhr-Universität Bochum. We thank Dr. Burkhard Mädler from Philips for his technical support with the multi-band sequence and Dr. Quoc Vuong, Elena Stebbings and Rohan Rao for critically reading, editing and commenting on the manuscript. We also would like to thank Dr. Aurelio Cortese and Dr. Ali Hummos for the valuable inputs during the revision of the paper.

## Author contributions

B.A.W., A.B., and B.P. designed the experiment. B.A.W. and M.V. performed the experimental data collection. B.A.W. carried out the data analysis. B.A.W., A.B., and B.P. discussed the results. B.A.W. and B.P.

wrote the paper with text, edits, and inputs from A.B. Both A.B. and B.P. supervised the work.

## Funding

## Competing interests
The authors declare no competing interests.
