## [Peer Review File · Nature Communications]

Human orbitofrontal cortex signals decision outcomes to sensory cortex during behavioral adaptationsREVIEWER COMMENTS

Reviewer #1 (Remarks to the Author):

This is an interesting study of a lateral orbitofrontal cortex in learning. Perhaps its key strength is that it not only reports activity in this area but it compares and relates it to activity in somatosensory cortex using insights derived from one of the author's previous rodent studies. Most of my points are very simple requests for clarification (points 3-7). Points 1 and 2 relate to the way that the study is framed. I do not think that the way that the authors talk about reversal is quite right and consistent with other results in the field. However, again, I think that this is relatively easy to update.

1. One point that has the potential to cause some confusion is the precise anatomical location that is being considered. This is consequential because, first, changing the way that this is discussed has may widen the impact that the study is likely to have. However, second, it may be a simple way to address what might otherwise be considered factual errors.

For example, from the very first page, the authors emphasize that there is broad agreement that the orbitofrontal cortex is concerned with reversal learning. For example, line 64 argues "The prefrontal cortex (PFC), more specifically the orbitofrontal cortex (OFC), has long been implicated in the ability to respond flexibly to obtain reward". However, the references that are cited do not make a strong case that the OFC is concerned with reversal learning. For example, the review by Murray and Rudebeck is particularly clear on this point. That review was influenced by an important paper that appeared in the previous year (*Nature Neuroscience*, 2014) published by the same authors in which they argued "We found that excitotoxic, fiber-sparing lesions confined to OFC in monkeys did not alter either behavioral flexibility, as measured by object reversal learning". In other words, Rudebeck and Murray contention is the opposite to the argument that they are cited in support of.

I feel that acknowledging this different perspective is important. This is especially the case given that the authors' admirable aim is, in part, to integrate our understanding of OFC across species. Part of the way out of the dilemma is that the region that seems to be important for a process integral to reversal learning is in the lateral part of orbitofrontal cortex. In macaques, this region lies in and lateral to the lateral orbitofrontal cortex and so it is outside the region typically referred to as 'OFC' (Chau et al., *Neuron*, 2015; Folloni et al., *Science Advances*, 2021). Rudebeck and Murray and colleagues (*Neuron*, 2017, *Nature Reviews Neuroscience*, 2018) even refer to it as ventrolateral prefrontal cortex. Perhaps the best way to deal with this issue is to refer to it as lateral OFC, or IOFC, as the authors do at some points or perhaps to point out that really the area is in between the foci of OFC and lateral prefrontal investigations and perhaps it is worth just referring to the area as area 47/12o (Petrides and Pandya, Mackey and Petrides, *EJN*, 2010; Petrides et al., *Cortex*, 2012; Neubert et al., *PNAS*, 2015). As far as I can see – a similar area is the one that is active in this manuscript in figure 2 and figure 5. By contrast the

area typically described as OFC, medial to the lateral orbitofrontal sulcus, seems to be less directly linked to the same processes.

2. One way to characterize the IOFC region that the authors discuss is to focus on its responsivity to rewards. However, this stands at odds with the emphasis on negative outcomes that is made in studies of IOFC (for example in some of the work of O’Doherty, Kringelbach, Rolls, and colleagues). One way to reconcile these perspective is if IOFC is concerned with crediting the outcome to a choice in a specific way so as to allow subsequent choices to be appropriate ones – for example by taking the choice again if it is rewarded and avoiding it if it is unrewarded (Chau et al., *Neuron*, 2015, Folloni et al., *Science Advances*, 2017). I realise that in the current study the authors show that a simple win-stay/lose-shift (WSLS) strategy that is based on just the last trial’s outcome does not explain behaviour well. Nevertheless, it is possible that the IOFC activity is specifically related to just rewards or alternatively that its activity is related to rewards and errors that have an impact on subsequent behaviour. The relative sizes of IOFC responses to reward and errors would then be related to the impact that each type of outcome has in driving adjustments of behaviour in any given task phase. I could imagine that the authors might not think so differently, however, it is easy to get the impression that they are arguing for a different idea – that OFC is linked to reward representation per se or value per se.

3. Line 204 – is the GLM that is described here, and the results illustrated in figure 3e, looking at activity time-locked to outcomes? I think that this is almost certainly the case but perhaps a brief mention in the text would be useful.

4. Line 218 emphasizes that figure 3c suggests IOFC encodes outcomes after a reversal rather than during initial learning. However, I think that the blue line at LN is really relatively high suggesting IOFC is active during initial learning. The authors argue that reference 4 argues that IOFC responses are most prominent during reversal rather than during initial learning but I do not think that is the case. I think that the work of all the authors of reference 4 is, in fact supportive of a role for IOFC in learning choice-outcome associations rather than in reversal per se.

5. Line 242. Are the representational similarity analyses (RSAs) time-locked to particular events – either to the stimulus occurrence or the outcome occurrence? Obviously, it makes sense to look at stimulus representation at the time of the stimulus and outcome representation at the time of the outcome. However, there is also evidence that orbitofrontal cortex neurons represent the choice taken at the time of the outcome even if they do not represent it at the time that the choice is actually taken (Tsujimoto et al., *J. Neurosci.*, 2009) perhaps because they are concerned with linking the association of the choice to the outcome.

6. P367-368. Could you clarify how the outcome-related activity was “later in the task” in somatosensory as opposed to orbitofrontal cortex? Do you mean later within a given trial or do you mean that it

occurred in later trials as opposed to earlier trials? Perhaps clarifying the argument and pointing the reader to the data that the authors are thinking of would be helpful.

Minor

7. Line 213 “who” should be “which”.

Reviewer #2 (Remarks to the Author):

In this paper, the authors performed fMRI experiments with a probabilistic tactile reversal learning task. The authors’ analyses focused on two ROIs, IOFC and S1, showing distinctive responses at different phases of learning, as well as potential interactions between these regions.

I found this paper potentially interesting. However, I am unsure if the data and the current analyses support their claims. More clarifications and careful analyses will be required to support their claims.

1. Behavioral analysis: the authors compared four models against behavioral data. However, I couldn’t find how parameters were fit to the behavior. Instead, it seems that the authors handpicked some parameters and compared simulated models with data. I am not sure if this is the case, but if so, parameters need to be fit to data to compare models.

2. Univariate analysis: The authors found prediction error signals in fMRI signals. However, according to the method section, the GLM contains the absolute value of prediction error instead of signed, full prediction error. I am very confused. Please clarify.

3. The GLM does not control outcome signals (reward or no reward). Prediction errors strongly correlate with outcomes. As a result, outcome correlation can be misclassified as a prediction error (Behrens et al., Nature). The authors need to either control outcome signals or show a positive correlation with outcomes and a negative correlation with expectations to test if the regions encode prediction errors.

4. Response changes across phases of learning: it is not clear what is actually computed in Figure 3. Is it a beta of GLM, or raw BOLD? Prediction error signals are presumably larger in LN and RN phases, which the authors show to be encoded in OFC. So OFC is just showing prediction error response?

5. RSA: I am not sure why the data’s RDMs are not shown in the paper. It would be more straightforward to see data.

6. Stimulus selective vs outcome selective: I was confused about why one expects selective stimulus coding in outcome onsets. Is this related to stimulus value coding? Then why not contrast it to action value coding?

7. PPI analysis. The GLM of PPI analysis does not include important regressors such as outcomes (reward - no reward) or prediction error. All the signal regressors need to be included because otherwise, what is shown here could be a signal correlation instead of a noise correlation.

Reviewer #3 (Remarks to the Author):

In their work, Wang and colleagues study the neural mechanisms by which S1 interacts with prefrontal structures (more specifically IOFC) during a probabilistic tactile learning task. This work can be seen as a follow-up of the original work by one of the co-authors (Banerjee et al 2020), but this time conducted in humans during fMRI, thus seeking integrative consistency across species. While I am enthusiastic about this kind of integrative work, I am less impressed by the specificity of the claims that the authors attempt to make during the presentation of the results both at the level of the modeling and the neural data. This in addition to the lack of clarity in the description of the models and results in several passages of the text. This substantially reduces my enthusiasm for the conclusions stated in this work. I expand on my comments below:

1) Modelling: In several passages of the manuscript, starting from the introduction and keywords, the authors claim that human participants employ a *Bayesian* strategy. Unfortunately, this claim has support neither with their experimental paradigm nor with their modeling approach. First, the behavioral paradigm implemented here has no statistical structure that the participants could learn to claim *Bayesian* belief updating. This is an associative probabilistic task with reversal where no parameter of the environment was systematically manipulated or controlled (e.g., Hazard rates) to be able to apply any form of optimal Bayesian inference.

In this work, the authors employ the HGF to fit their choice data. But employing HGFs and claiming that because it fits the data better, the participants adopt Bayesian inference, is not correct. Note that the RW can also be derived using Bayesian formulations, and in fact, any statistical model can be implemented using Bayesian formulations. But this does not mean that participants invert a Bayesian model based on a learned hierarchical structure of the environment or a given context. This is without mentioning that the HGF employs quite some variational approximations to make the model tractable which then makes the normative approach questionable (e.g., see Piray and Daw 2020). I am not someone who is against Bayesian models (actually quite the opposite), but claims of Bayesian inference need the design of specific behavioral paradigms and the development of the corresponding Bayesian inversion models (e.g., see Ma, 2012 TiCS).

2) Model specification and fits: The explanation of the model implementation, in particular the HFG is incomplete and difficult to follow without reading the original work. For instance, before equation 8 the authors introduce ϵ , but it is not applied in any of the subsequent equations. Also is ϕ a free parameter? How is δ exactly defined? Additionally, there is no information on how the models are fit to the data. Were the fits being done for each participant? What were the free parameters? (in the case of the HGF, what is with ω ?, etc. or which were the actual free parameters, were some parameters set to fix values, in case yes, then how these were selected). In the main text line 137, where does the sentence: “particular set of parameters” come from? Moreover, it will be important to show the quality of the model fits. In Figure 1 the only information regarding this is shown in panel e but it is not clear if this corresponds to the average data with the average model predictions of the model fit individually for each participant. It would be more convincing to show how learning behavior as shown in panel d is captured by the model. Also, please provide a table (main text or supplement) with the average parameters \pm s.d. of the parameters fit for each model that would allow qualitative reproduction of the predictions.

3) Results section starting on line 184: Please mention what exact parameter of the HGF model presented in the methods section was used to compute the PEs: was this the choice prediction error? Or is it one of the other higher-level PEs (ϵ) in the HFG? If it is the choice prediction error, it is unclear to me why a well-defined structure playing a role in choice PEs that is present in basically all fMRI studies such as the ventral striatum (including the original work employing HFGs, Iglesias et al, 2013 Neuron) does not show up in this work? If it is not choice prediction error but a higher level prediction error, does this PE match the type of PE studied in Banerjee et al 2020? I presume not as I think in Banerjee et al 2020 the authors studied choice PEs in. A key reason why this is critical to clarify is that both types of PE appear to make different contributions at the neural circuit level (again see: Iglesias et al, 2013 Neuron). In this work, it is also not clear whether both levels of PE from the HGF were introduced as regressors in the same GLM. This is essential as not doing so might bias the results and their interpretation.

4) Results Figure 2: From the description of the main text and the figure legend, it is unclear whether the brain “activations” and activities (in panel c) have any relation to the HFG model or they are mere “activations” relative to baseline. If they are baseline relative activations, it is unclear why for LN trials S1 would have activity with values near zero. This is a sensorial area and it is not expected to have activations at around baseline for the first trials. Moreover, this finding goes against the common observation of BOLD adaptation over time, and this has been particularly evidenced in sensory areas. How do the authors explain this discrepancy?

5) Results Figure 3: It is not entirely convincing that the structure shown in Figure 3a is IOFC, it looks more like the anterior insula. Additionally, are the clusters shown in the two brain slices in Figure 3b the same? If the dimensions of the two slices were kept on the same scale, then the clusters shown in the two cuts appear to be from different cortical locations.

6) Discussion and interpretation: It is well taken that the authors intend to build a translational story between the rodent work and this one. However, the question here is whether the S1 results are specific to interactions with IOFC. One of the main reasons I mention this is that there were many other prominent structures such as MFG, PPC, SMA that apparently did not require small volume correction in the first set of analyses. Therefore, the question is whether the authors can assure that the IOFC results are specific to this structure. Moreover, structures such as MFG, PPC or SMA are stronger candidates to modify S1 function due to closer proximity and also likely monosynaptic connections to S1. It was the case in rodents, but here it is important to question whether it has been evidenced that the IOFC areas localized in this study have indeed direct connections to S1? In case not, the motivation and discussion around this precise IOFC to S1 anatomical interaction might be quite a stretch, and even more in light of the finding in this study that more proximal structures appear to also be involved in the mechanisms studied in this task.

7) RSA analyses: The times between stimulus and outcome appear to be extremely short an average of around two seconds or less appears insufficient for the BOLD response to decay sufficiently.

Responses to reviewers' comments

We would like to thank the editor and reviewers for giving us the opportunity to respond to the valuable comments made on the manuscript. Based on these comments, the manuscript has undergone substantial revision. We would like to thank each reviewer for their constructive criticism, which has helped to strengthen the conclusion of our manuscript.

Here, we summarize an overview of major changes to the manuscript:

1. We added model simulations and optimized model parameters for the comparison of simulated and observed behavioral responses. We also validated the superiority of the winning model and showed how this model captures learning behavior. These additional analyses have helped to solidify the conclusion that the HGF model provides the best fit for participants' performance (see our responses to comments from Reviewer 2 and 3; see also new Supplementary Fig.1 and 2 and Supplementary Table1 and 2).
2. We re-analyzed the fMRI univariate GLM models adding additional regressors of outcomes (reward and error) to control their potential influence on the activity of IOFC and S1 after the reversal. The updated models resemble the original results suggesting that the responses from IOFC and S1 after reversal were not just affected by outcome signals (updated Fig. 3). We also added the same regressors to the PPI analysis, which, like the univariate GLM, resembled the original findings suggesting that the strengthened connectivity between the outcome-selective IOFC and ipsilateral S1 (updated Fig. 6), immediately after a reversal, was also not specifically related to outcomes. For details, please see our responses to comments from Reviewer 2.
3. We added the data RDM to the RSA results in the updated Fig. 4. We also run an analogous RSA analysis time-locked to the stimulus presentation to test the stimulus-selectivity of S1 (see our responses to comments from Reviewer 1 and 2; see also new Supplementary Fig.5).
4. We substantially revised the method section offering more detailed and comprehensive descriptions, especially with respect to the HGF model, based on the comment from Reviewer 3. Furthermore, we provided a more detailed discussion on the precise anatomical topography of the area we assigned to the IOFC inspired by the comments from Reviewer 1.

Below, please find our point-by-point responses to each reviewer's comments. Sentences in black correspond to the reviewer's comments, followed by our responses in blue. We have highlighted all changes in the original manuscript.

REVIEWER COMMENTS

Reviewer 1 (Remarks to the Author):

This is an interesting study of a lateral orbitofrontal cortex in learning. Perhaps its key strength is that it not only reports activity in this area but it compares and relates it to activity in somatosensory cortex using insights derived from one of the author's previous rodent studies.

Most of my points are very simple requests for clarification (points 3-7). Points 1 and 2 relate to the way that the study is framed. I do not think that the way that the authors talk about reversal is quite right and consistent with other results in the field. However, again, I think that this relatively easy to update.

We thank the reviewer for her/his encouraging comments, and for highlighting the strengths of our study.

1. One point that has the potential to cause some confusion is the precise anatomical location that is being considered. This is consequential because, first, changing the way that this is discussed has may widen the impact that the study is likely to have. However, second, it may be a simple way to address what might otherwise be considered factual errors.

We thank the reviewer for this important point. We acknowledge that the precise assignment of the observed effects to the underlying anatomy is very critical which already caused confusion about the functional topography of brain structures across species in the past. OFC is one of the prominent structures in the cross-species literature on cognitive flexibility where the assignment of effects, especially to its lateral parts, caused confusions. In order to avoid these confusions, we introduced the function of lateral OFC specifically in credit assignment of outcomes and discussed the potential discrepancy in its anatomical assignments especially in cross-species studies. Please see our detailed responses and revisions below.

For example, from the very first page, the authors emphasize that there is broad agreement that the orbitofrontal cortex is concerned with reversal learning. For example, line 64 argues “The prefrontal cortex (PFC), more specifically the orbitofrontal cortex (OFC), has long been implicated in the ability to respond flexibly to obtain reward”. However, the references that are cited do not make a strong case that the OFC is concerned with reversal learning. For example, the review by Murray and Rudebeck is particularly clear on this point. That review was influenced by an important paper that appeared in the previous year (Nature Neuroscience, 2014) published by the same authors in which they argued “We found that excitotoxic, fiber-sparing lesions confined to OFC in monkeys did not alter either behavioral flexibility, as measured by object reversal learning”. In other words, Rudebeck and Murray contention is the opposite to the argument that they are cited in support of.

We apologize for the inaccuracy in citing the work by Murray and Rudebeck, who argued that excitotoxic, fiber-sparing lesions of both medial and lateral OFC have no effect on the object reversal learning, but on choosing objects or an action based on the expected outcome value in the devaluation task. Based on these findings, they concluded that OFC is necessary for updating the value of specific stimulus-outcome associations, not simply tracking the presence, absence, or likelihood of a reward.

We revised and extended the argument in the introduction accordingly (**Lines 35-44**):

‘Within the elaborate frontal cortical areas involved in flexible decision-making, the orbitofrontal cortex (OFC) has been one of the most intensively studied structures and is known to have widespread connectivity to sensory areas, as well as to cortical and subcortical areas related to memory, learning, and attention (Cavada et al. 2000; Kringelbach and Rolls 2004). OFC is specifically implicated in choosing objects or an action

based on the expected outcome value and updating the value of different stimulus-outcome associations (Rudebeck et al. 2013; Rudebeck and Murray 2014). Compared to medial orbitofrontal cortex (mOFC) which encodes the reward value to support choices, lateral OFC (IOFC) is relatively more specialized for assigning credit for both rewards and errors to specific stimulus choices, emphasizing the IOFC's role in learning about the values of options (Rushworth et al. 2011).'

I feel that acknowledging this different perspective is important. This is especially the case given that the authors' admirable aim is, in part, to integrate our understanding of OFC across species. Part of the way out of the dilemma is that the region that seems to be important for a process integral to reversal learning is in the lateral part of orbitofrontal cortex. In macaques, this region lies in and lateral to the lateral orbitofrontal cortex and so it is outside the region typically referred to as 'OFC' (Chau et al., Neuron, 2015; Folloni et al., Science Advances, 2021). Rudebeck and Murray and colleagues (Neuron, 2017, Nature Reviews Neuroscience, 2018) even refer to it as ventrolateral prefrontal cortex. Perhaps the best way to deal with this issue is to refer to it as lateral OFC, or IOFC, as the authors do at some points or perhaps to point out that really the area is in between the foci of OFC and lateral prefrontal investigations and perhaps it is worth just referring to the area as area 47/12o (Petrides and Pandya, Mackey and Petrides, EJN, 2010; Petrides et al., Cortex, 2012; Neubert et al., PNAS, 2015). As far as I can see – a similar area is the one that is active in this manuscript in figure 2 and figure 5. By contrast the area typically described as OFC, medial to the lateral orbitofrontal sulcus, seems to be less directly linked to the same processes.

We thank the reviewer for this important discussion about the variability in anatomical assignments throughout the corresponding literature. Motivated by the reviewer's comment and the references provided, we added a new paragraph to the Discussion section to make the reader aware of this problem (**Lines 358-372**):

'The topographic assignment of the 'IOFC' region we identified in the context of outcome-selectivity is approximate and should be further confirmed. In the literature, especially in macaque monkeys, who present a comparable prefrontal architecture as humans (Neubert et al. 2015; Petrides et al. 2012), diverse anatomical nomenclatures were used. Rudebeck et al., for instance, defined a broad orbital-lateral prefrontal area named ventrolateral prefrontal cortex (Murray and Rudebeck 2018; Rudebeck et al. 2017), which was involved in credit assignment, and which encompasses the monkey equivalent of the outcome-selective region we identified between the foci of OFC and lateral prefrontal cortex. Folloni et al. also identified credit assignment-related functions in a comparable albeit smaller ventrolateral prefrontal region (Folloni et al., 2021), which they referred to as 47/120 according to Brodmann's cytoarchitectural brain atlas, and which substantially overlaps with the regions we found in the context of outcome-selectivity (see Fig. 2 and Fig. 5). In our study, we referred to this outcome-selective prefrontal region as IOFC, but the corresponding neural activity we found in this area may have slightly extended into adjacent areas, such as the dorsolateral prefrontal cortex. We cannot rule out that these areas may have also contributed to generating decision outcome signals.'

2. One way to characterize the IOFC region that the authors discuss is to focus on its responsivity to rewards. However, this stands at odds with the emphasis on negative outcomes that is made in studies of IOFC (for example in some of the work of O'Doherty, Kringelbach, Rolls, and

colleagues). One way to reconcile these perspective is if IOFC is concerned with crediting the outcome to a choice in a specific way so as to allow subsequent choices to be appropriate ones – for example by taking the choice again if it is rewarded and avoiding it if it is unrewarded (Chau et al., Neuron, 2015, Folloni et al., Science Advances, 2017). I realise that in the current study the authors show that a simple win-stay/lose-shift (WSLS) strategy that is based on just the last trial's outcome does not explain behaviour well. Nevertheless, it is possible that the IOFC activity is specifically related to just rewards or alternatively that its activity is related to rewards and errors that have an impact on subsequent behaviour. The relative sizes of IOFC responses to reward and errors would then be related to the impact that each type of outcome has in driving adjustments of behaviour in any given task phase. I could imagine that the authors might not think so differently, however, it is easy to get the impression that they are arguing for a different idea – that OFC is linked to reward representation per se or value per se.

We apologize for the confusing discussion about IOFC as a region that specifically responds to reward. We considered all type of trials including both reward and error trials (HIT, CR, FA and MISS) in the current study. We concluded that the IOFC encodes deviations from expected outcome value after a rule switch by tracking both positive and negative outcomes. Based on the revised GLM models, extended by reward/error regressors, we now emphasize throughout the revised manuscript that IOFC activity is related to signaling outcome values but not specifically rewarding outcomes.

To test the reviewer's well-justified hypothesis that responses from IOFC may rather reflect the outcome value of both reward and error, we performed two additional analyses:

(1) In order to test the potential influence of unexpected reward and error outcomes on credit assignment, we separately analyzed the BOLD activity in IOFC for reward (HIT, CR) and error trials (FA, MISS) across the four phases of the learning task (Fig. R1). We found IOFC activity in both, unexpected reward (HIT and CR) and unexpected error (FA and MISS) trials immediately after the reversal, suggesting that the activity in IOFC was not specifically related to reward or no reward outcomes (within-subject repeated measures ANOVA: no significant interaction between phases and types of trials ($F(1,31) = 0.95, p = 0.42$) and the main effect of types of trials ($F(1,31) = 0.74, p = 0.53$); only a significant main effect of phases ($F(1,31) = 17.87, p = 0.00019$)). Please see also our responses to comments 3 and 4 by Reviewer 2.

Fig. R1 (Supplementary Fig. 3 in the main text). The BOLD signals in IOFC across the four key learning phases for the four types of trials (HIT, CR, FA, MISS) respectively. The error bars indicate the SEM.

(2) We also re-analyzed our entire fMRI models extended by additional regressors accounting for rewards and errors on a trial-by-trial basis. These additional outcome regressors, however, did not reveal any significant effects on the responses of IOFC, additionally emphasizing that activity in IOFC was not specifically related to rewards or errors. See also our response to comment 3 by Reviewer 2 and the new results we present there.

We thank the reviewer for this important hint which has helped to specify our fMRI analyses further and, hence, corresponding data interpretations.

3. Line 204 – is the GLM that is described here, and the results illustrated in figure 3e, looking at activity time-locked to outcomes? I think that this is almost certainly the case but perhaps a brief mention in the text would be useful.

Yes, the reviewer is correct. The GLM analysis and the result shown in Figure 3 are time-locked to the onset of outcomes. We added this important information to the method section, which reads as follows (**Lines 174-176**):

'To examine whether activity in these two regions related to outcomes following reversals and re-learning, we applied two independent general linear model (GLM) analyses time-locked to the onset of the outcome.'

4. Line 218 emphasizes that figure 3c suggests IOFC encodes outcomes after a reversal rather than during initial learning. However, I think that the blue line at LN is really relatively high suggesting IOFC is active during initial learning. The authors argue that reference 4 argues that IOFC responses are most prominent during reversal rather than during initial learning but I do not think that is the case. I think that the work of all the authors of reference 4 is, in fact supportive of a role for IOFC in learning choice-outcome associations rather than in reversal per se.

We thank the reviewer for this comment. We found increased activity in IOFC during both naïve phases (LN and RN), and not just RN. In terms of RN, we argued that IOFC encodes the deviations from expected outcome values and represents credit assignment of unexpected outcomes to specific stimulus-response associations. We cited Rushworth et al., because of their argumentation that IOFC, as compared to vmPFC and mOFC, specifically supports the assignment of credit for outcomes to specific stimulus choices through reactivating a representation of the stimulus choice that had just been made at the time when the reward was received or when the reward was absent due to an error (Rushworth et al. 2011).

Combining the response to comment 2, we rephrased the corresponding paragraph to focus on the role of IOFC during RN in credit assignment of both, unexpected rewards and errors, to specific stimulus-response associations (**Lines 192-204**):

'During RN, we again found transient but large IOFC responses to unexpected outcomes, which decreased as participants re-learned the task during RE (Fig. 3c). To test the potentially differential influence of appetitive and aversive outcomes, we analyzed IOFC activity in trials with rewards (HIT, CR) and errors (FA, MISS) for each of the four learning phases separately (Supplementary Fig. 3). IOFC responded to both, unexpected reward and unexpected error trials immediately after the reversal (RN, within-subject repeated measures ANOVA: no significant interaction between phases and types of trials ($F(1,31) = 0.95$, $p = 0.42$) and the main effect of types of trials ($F(1,31) = 0.74$, $p = 0.53$); only a significant main effect of phases ($F(1,31) = 17.87$, $p = 0.00019$), suggesting that IOFC encodes deviations from expected outcome values after a reversal to assign credit to specific stimulus-response associations, irrespectively of whether they have been rewarded or not (Rushworth et al. 2011).'

For LN (i.e., initial learning phase), we also found increased activity in IOFC, as the reviewer correctly pointed out. To statistically test this, we performed a GLM analysis comparing LN and LE trials. We observed significantly enhanced BOLD signals in bilateral IOFC during the initial learning (Right: $x = 38$, $y = 56$, $z = 0$, $t(31) = 5.70$, $p_{\text{FWE-SVC}} = 0.005$; Left: $x = -42$, $y = 48$, $z = 0$, $t(31) = 4.87$, $p_{\text{FWE-SVC}} = 0.037$, Fig. R2). This finding stands in discrepancy to cellular imaging in our mice study (Banerjee et al. 2020), in which IOFC engagement was observed in the naïve phase after the reversal (RN), but not in the naïve phase during initial learning (LN). In our original manuscript, we interpreted this discrepancy in the context of the probabilistic nature of the reversal learning task, which stands in contrast to the deterministic task applied to mice. In our revision, we now emphasize that human IOFC seems to encode deviations from expected outcomes also in the initial learning phase, suggesting a supportive role of IOFC in the initial

learning of probabilistic cue-outcome associations. In support of these conclusions, we cite lesion studies that showed that damage to lateral OFC was particularly associated with decreased adaptation cycles during the reversal phase of a simple deterministic reversal learning task (Fellows and Farah 2003), but in a more challenging probabilistic environment, IOFC damage disrupted both, initial and reversal learning (Tsuchida, Doll, and Fellows 2010).

Fig. R2. The IOFC activity engaged in the initial learning phase (LN>LE). T-maps are displayed at $p < 0.001$, uncorrected, for display purpose. The activity of IOFC is significant at $p < 0.05$, small-volume FWE corrected.

In the revised results, we now emphasize that IOFC was significantly activated also during LN compared to LE and that this finding stands in discrepancy to our recent rodent study (**Lines 204-210**):

'In the initial learning phase (LN > LE), IOFC also encoded deviations from expected outcomes, suggesting a supportive role of IOFC in the initial learning of probabilistic cue-outcome associations. Notably, only humans, and not mice, showed responses in IOFC during LN, which may be interpreted in the context of the task design, which was probabilistic for humans, but deterministic for mice (see Discussion for further details). Further mechanistic investigations in mice under probabilistic demands are required.'

5. Line 242. Are the representational similarity analyses (RSAs) time-locked to particular events – either to the stimulus occurrence or the outcome occurrence? Obviously, it makes sense to look at stimulus representation at the time of the stimulus and outcome representation at the time of the outcome. However, there is also evidence that orbitofrontal cortex neurons represent the choice taken at the time of the outcome even if they do not represent it at the time that the choice is actually taken (Tsujimoto et al., J. Neurosci., 2009) perhaps because they are concerned with linking the association of the choice to the outcome.

Yes, the RSA analysis in our study was time-locked to the onset of outcomes. Our hypothesis for both the RSA and PPI analysis is that, at the time of outcome presentation, IOFC assigns credit for unexpected outcomes after reversal to a given stimulus (Go-tactile pattern or NoGo-tactile

pattern) through signaling the outcome values to the sensory cortex. In line with our hypothesis, previous studies revealed compelling evidence for outcome-related responses in primary sensory cortices during outcome presentation (Brosch, Selezneva, and Scheich 2011; Pleger et al. 2008, 2009; Poort et al. 2015) – a signal which is thought to support the assignment of rewarding outcomes to sensory stimuli. Our results of outcome-selective representations in S1 at the time of outcome presentation together with the strengthened connectivity between outcome-selective IOFC and S1 suggests that these regions jointly support the assignment of outcome values to corresponding stimulus-response associations. In this context, it is important to note that our task is not a pure stimulus-outcome learning task. Instead, it is a stimulus-action-outcome learning task where outcome values are rather assigned to the association between the stimulus and the Go/NoGo response than to the stimulus alone.

Motivated by the reviewer's comment and in line with our recent mice study, we performed an additional RSA analysis, time-locked to the stimulus presentation. To account for stimulus-selectivity in S1, we expected a strong similarity of each stimulus between before reversal (LE) and after reversal (RN or RE) (Fig. R3a). Fig. R3b shows the representational dissimilarity matrix (RDM) of the response pattern from S1 for the immediate effect of reversal (LE->RN) and the adaptation after re-learning (LE->RE). In this context, we only found weak stimulus encoding in S1 (see Fig. R3c). We additionally tested contralateral and ipsilateral S1 separately, but the results did not differ. We interpret these findings as follows: At the time of stimulus presentation both stimuli share the same representation in S1 due to common sensory features, such as the same stimulation intensity and the same number of stimulating pins (i.e., 4).

We added this additional finding to the result section together with a new Supplementary Fig. 5. The new RSA analysis is discussed as follows:

'An analogous stimulus-selective RSA analysis for S1, with the onset placed on the time of stimulus presentation, revealed no evidence for different representations of the two alternative stimuli. This suggests that both tactile stimuli share the same S1 representation due to common sensory features, such as the same stimulation intensity and the same number of stimulating pins (Supplementary Fig. 5)' (Lines 262-266).

'During stimulus presentation, S1 presented a rather weak stimulus-selective response, but during the outcome phase, we found strong stimulus- and outcome-selective S1 responses. These representations in S1 are important for linking the outcome value to the corresponding sensory stimulus, which is in line with previous evidence suggesting that outcome feedback is associated with activity in sensory areas involved in stimulus processing during outcome presentation, even in the absence of concurrent sensory stimulation (Brosch et al. 2011; Pleger et al. 2008, 2009; Poort et al. 2015). One way the brain might perform this operation is to encode and transmit a 'teaching' signal, based on reward or error outcomes, to sensory regions involved in stimulus processing (FitzGerald, Friston, and Dolan 2013; Roelfsema, van Ooyen, and Watanabe 2010).' (Lines 373-381).

a Stimulus-selective model: RDM

b Data: RDM

c

Fig. R3 (Supplementary Fig. 5 in the main text). The RSA for stimulus-selectivity in S1, time-locked to stimulus presentation. a. Description of the stimulus-selective model and the corresponding representational dissimilarity matrix (RDM). b. The RDM for stimulus-selective responses from S1 for both LE->RN and LE->RE. c. The comparison between the mean 'similar' (black elements in model RDMs) and mean 'dissimilar' (white elements in model RDMs) using a one-sided Wilcoxon signed-rank test together with permutation tests. n.s. indicates non-significance.

6. P367-368. Could you clarify how the outcome-related activity was “later in the task” in somatosensory as opposed to orbitofrontal cortex? Do you mean later within a given trial or do you mean that it occurred in later trials as opposed to earlier trials? Perhaps clarifying the argument and pointing the reader to the data that the authors are thinking of would be helpful.

We thank the reviewer for this important question. According to the RSA and PPI results, shown in Figure 5 and Figure 6 in the main text, we identified outcome-related IOFC activity in RN trials, directly after the reversal. This IOFC activity was strongly related to outcome-selective S1 signals that, however, became significant at a later time point in the task, during the re-learning phase after the reversal (RE). Based on this time order, we assumed that the outcome-selective ipsilateral S1 activity during the re-learning phase (RE) depends on OFC signals that reflect outcome-selectivity much earlier in the task, namely immediately after switching the learning rule (RN). We were also conceptually guided by our earlier study (Banerjee et al. 2020) that used

pharmacogenetic manipulations to block IOFC activity during RN that affected response remapping in S1 in RE.

For a better description of the logic behind our assumption, we included a new schematic figure (Fig. R4) of the stimulus and outcome selectivity in S1 and IOFC before and after the rule switch. Based on these results, we conclude that the IOFC assigns credit for unexpected outcomes to sensory stimuli after the rule switch (RN) through signaling outcome values to the sensory cortex, which, in turn, supports later re-learning (RE).

Fig. R4 (Supplementary Fig. 6 in the main text). The schematic describes the representation of stimulus and outcome in S1 and IOFC after reversal.

We have rewritten the corresponding sentences in the Discussion to clarify our conclusion as follows (**Lines 331-339**):

‘However, according to the RSA and PPI results shown in Fig. 5 and Fig. 6, we found that outcome-related IOFC activity immediately after the rule switch (i.e., reversal) closely related to the outcome-selective ipsilateral S1 signal, that, however, occurred later in the task, after re-learning (RE). In light of this time order, it is more likely that the outcome-selectivity in IOFC is responsible for shaping outcome-selectivity in ipsilateral S1, than the other way around (Supplementary Fig. 6). Our findings extend observations in mice, suggesting that IOFC is involved in assigning credit for unexpected outcomes to specific stimulus-response associations through signaling the outcome values to the sensory cortex, which concurrently results in behavioral adaptation (Banerjee et al., 2020).’

Minor

7. Line 213 “who” should be “which”.

This has been corrected in the revised manuscript.

Reviewer 2 (Remarks to the Author):

In this paper, the authors performed fMRI experiments with a probabilistic tactile reversal learning task. The authors' analyses focused on two ROIs, IOFC and S1, showing distinctive responses at different phases of learning, as well as potential interactions between these regions.

I found this paper potentially interesting. However, I am unsure if the data and the current analyses support their claims. More clarifications and careful analyses will be required to support their claims.

We thank the reviewer for considering our work potentially interesting. Following the comments of the reviewer, we have substantially revised our analyses. Below, we describe how we addressed every single point. We thank the reviewer for his/her thoughtful comments that have helped to improve our analyses and strengthen our claims. We hope the reviewer agrees with these changes.

1. Behavioral analysis: the authors compared four models against behavioral data. However, I couldn't find how parameters were fit to the behavior. Instead, it seems that the authors handpicked some parameters and compared simulated models with data. I am not sure if this is the case, but if so, parameters need to be fit to data to compare models.

We thank the reviewer for this important comment.

Our first intention was to apply default parameters from the literature, constrained to human-like overall performance. Both the RW and three-level HGF model have been extensively studied using probabilistic learning tasks (Iglesias 2013; Vossel et al. 2014, 2015; Wang and Pleger 2020). Considering the fact, that our task is a reversal learning task with low volatility, for which we applied the two-level and not three-level HGF model, we agree with the reviewer, that we should rather explore the set of parameter values which better account for participants' real behavior.

Therefore, we first simulated behavior across a range of parameter settings to determine how the model-independent measures change with different parameter settings. Specifically, we plotted the $p(\text{correct})$ of simulated responses as a function of free parameter values for each model (random: $b=[0\ 1]$; WSLS: $\epsilon=[0\ 1]$; RW: perceptual model parameter $\alpha=[0\ 1]$, response model parameter $\beta=[1\ 20]$; HGF: perceptual model parameter $\omega=[-10\ -1]$, response model parameter $\zeta=[1\ 20]$). The simulation results are shown in Fig. R5.

For the Random and WSLS model, the choice of free parameters only marginally affected the probability of correct responses. Note that the WSLS model equals the Random model when $\epsilon=1$. In order to characterize the behavioral performance for the WSLS model, ϵ should be as small as possible. For the RW model, the 'optimal' learning rate α , that maximizes $p(\text{correct})$, ranges between 0.1 and 0.3. The 'optimal' step size ω in the HGF model ranges between -4 and -2. For both the RW and HGF model, the application of the response model parameters (β in the RW model, ζ in the HGF model) did not differ from the application of 'optimal' perceptual model parameters - it just changed overall model performance.

Fig. R5 (Supplementary Fig. 1 in the main text). Four model simulations (500 per parameter setting) show how the free parameters influence behavior

Based on these simulations, we optimized the free parameters for the four models. For the Random and WSLs model, we applied $b=0.5$ and $\epsilon=0.05$, as the same free parameters as before. For the RW and HGF model, we calculated the 'Bayes optimal' perceptual parameters for a given input sequence μ using the function in the HGF toolbox. 'Bayes optimal', in this context, are those parameters that produce the least cumulative Shannon surprise for a given input sequence μ . This means that an agent, using this parameter setting, would experience the least possible surprise when exposed to the given inputs in the given perceptual model. The 'Bayes optimal' perceptual parameters for both the RW and HGF models are listed in the following Table R1:

Table R1 (Supplementary Table 1 in the main text). The Bayes optimal parameters for the perceptual model in RW and HGF

	α in RW	ω in HGF		α in RW	ω in HGF
Block1	0.143	-3.25	Block7	0.147	-3.03
Block2	0.137	-3.61	Block8	0.141	-3.19
Block3	0.141	-3.33	Block9	0.142	-3.38
Block4	0.150	-2.88	Block10	0.138	-3.52
Block5	0.140	-3.65	Block11	0.147	-2.99
Block6	0.143	-3.04	Block12	0.137	-3.49

The free parameter of the response model was chosen based on both, the simulations and the range of parameters obtained from fitting the real behavioral performance. The detailed fit parameter values (means and standard error) for the RW and HGF model were shown in Table R2 (please also refer to comment 2 from Reviewer 3). Finally, $\beta=5$ for the RW model and $\zeta=10$ for the HGF model were used to simulate the response. Combining the Bayes Optimal perceptual parameters and fit response parameter values ($\beta=5$ for RW model and $\zeta=10$ for HGF model), we simulated the response and plotted $p(\text{staying})$ as a function of the outcomes (correct or wrong) of the previous trial for each of the four models. In agreement with our previous findings, participants' performance was better fitted by both, the RW and the HGF model (Fig. R6). To determine which of these two models better captures the behavioral data, we compared relative log-model

evidence between the HGF and RW model on the individual level (HGF was superior in 28 out of 32 participants) and group level (posterior probabilities: 0.95; exceedance probability = 1.00). In order to further validate the winning HGF model, we also plotted the averaged behavioral performance to show how real learning behavior is captured by the simulated responses from the HGF model (See Fig. R13 in the comment 2 from Reviewer 3).

Fig. R6 (Updated Fig. 1f-h in the main text). The model comparison. Left, the probability of repeating a decision, $p(\text{staying})$, as a function of the outcome of the previous trial for simulated responses, derived from the four model simulations, and the observed behavior. Right, the Bayesian Model Selection in group and individual level.

We show these model simulation and fitting results in the new Supplementary Figures 1 and 2, and updated Fig.1 in the main text.

Additionally, we explicitly explain how we choose these parameter values in the revised manuscript (**Lines 116-132**).

'First, we simulated behavior across a range of parameter settings to determine how the model-independent measures change with different parameters (Supplementary Fig. 1). Then, we tested one measure that captures fundamental aspects of the flexible decision process based on prior experience: the probability of repeating a decision, $p(\text{staying})$. To this end, we first estimated the free parameters by fitting them to the real behavioral performance (Supplementary Table 1). The choice of free response parameters (β in RW and ζ in HGF) was based on the simulation results and in the range of parameters obtained from the estimation of model fitting. For the free perceptual parameters (α in RW and ω in HGF), we calculated the 'Bayes optimal' values for a given input sequence μ (Supplementary Table 2). Based on these simulations and model fitting results, we finally simulated the responses and calculated $p(\text{staying})$ using the four different models with a particular set of parameters (M1: $b = 0.5$; M2: $\varepsilon = 0.05$; M3: $\beta = 5$; M4: $\zeta = 0.5$; for more

details, please refer to the Methods section). These simulation parameters also match the typical parameter values used in the literature to constrain overall performance (Wilson and Collins, 2019). We repeated the simulation process 1000 times and plotted $p(\text{staying})$ as a function of the outcomes (correct or wrong) of the previous trial for each of the four models (Fig. 1f).'

2. Univariate analysis: The authors found prediction error signals in fMRI signals. However, according to the method section, the GLM contains the absolute value of prediction error instead of signed, full prediction error. I am very confused. Please clarify.

We apologize for this confusion. The outcome prediction error was defined as the difference between the actual outcome after the response to a given stimulus (i.e., stimulus1->'Go') and its a priori probability. In the contingency space, the sign of this outcome prediction error depends on the arbitrarily chosen coding of a binary input, in our case, $x_1^{(t)} = 1$ for stimulus1->'Go'/stimulus2->'NoGo' and $x_1^{(t)} = 0$ for stimulus1->'NoGo'/stimulus2->'Go'. Therefore, we used an unsigned outcome PE (i.e., the absolute value of original outcome prediction error), bounded between 0 and 1. Notably, the unsigned prediction error corresponds to Bayesian surprise and is equivalent to the prediction error we referred to in the rodent study (Banerjee et al. 2020).

We have clarified this point in the updated manuscript as follows (**Lines 587-592**):

'Notably, the sign of outcome prediction error in contingency space depends on the arbitrarily chosen coding of a binary input (in our case, the assignment of $x_1^{(t)} = 1$ for the stimulus1->'Go'/stimulus2->'NoGo' and $x_1^{(t)} = 0$ for the stimulus1->'NoGo'/stimulus2->'Go'). In this study, we used the unsigned outcome prediction error (i.e., absolute value) that corresponds to Bayesian surprise (Ide et al. 2013; den Ouden et al. 2009) and which is equivalent to the prediction error we investigated in our previous rodent study (Banerjee et al. 2020).'

3. The GLM does not control outcome signals (reward or no reward). Prediction errors strongly correlate with outcomes. As a result, outcome correlation can be misclassified as a prediction error (Behrens et al., Nature). The authors need to either control outcome signals or show a positive correlation with outcomes and a negative correlation with expectations to test if the regions encode prediction errors.

We greatly appreciate the reviewer's comment which has helped to optimize the GLM design and to better control for confounding variables.

In the new GLM, we included the reward/error outcomes as two additional regressors. After we update the GLM design matrix, we re-analyzed the model to assess responses, (1) related to the outcome prediction error, and (2) engaged in the reversal and re-learning phases (i.e., contrasting LE and RN, LE and RE). The updated results are shown in Fig. R7 and R8.

Fig. R7 (Updated Fig. 2 in the main text): fMRI activity related to the outcome PE. T-maps are displayed at $p < 0.001$, uncorrected, for display purpose. The activity of IOFC was significantly correlated with PE ($p < 0.05$, small-volume FWE corrected).

Fig. R8 (Updated Fig. 3a and 3b in the main text): Distinct engagement of lateral OFC and S1 after reversals and during re-learning, respectively. T-maps are displayed at $p < 0.001$, uncorrected, for display purpose. Both the activity of IOFC and S1 were significantly higher after reversal (RN, RE for IOFC and S1 respectively) compared to LE ($p < 0.05$, small-volume FWE corrected).

The new results greatly resemble those we presented in the original manuscript, suggesting that IOFC activity is related to signaling outcome values but not specifically rewarding outcomes (see also comment 2 by Reviewer 1). Despite minor differences between both results, such as the size of clusters or slight variations of significance value, we found the same brain regions related to PE, and the IOFC and S1 engagement immediately after reversal and re-learning, respectively.

We now report the details of updated results in the main text as follows:

'the responses in IOFC were significantly correlated with outcome PE ($x = 40$, $y = 48$, $z = -2$, $t_{(31)} = 4.94$, $p = 0.04$, family-wise error (FWE) peak-level corrected for multiple comparison using small-volume correction (SVC), Fig. 2).' (Lines 160-163)

'First, by comparing LE and RN trials, we observed significantly enhanced BOLD signals in right IOFC immediately after switching the stimulus-response association (two clusters: $x = 38, y = 24, z = -8, t_{(31)} = 7.17, p_{FWE-SVC} < 0.001$; $x = 36, y = 50, z = 0, t_{(31)} = 5.15, p_{FWE-SVC} = 0.02$, Fig. 3a). Based on the Automated Anatomical Labeling (AAL) atlas, both clusters were assigned to the IOFC, but the latter cluster was closer to the IOFC cluster that was associated with the encoding of the prediction error (Fig. 2). Second, by comparing LE and RE trials, we identified bilateral S1, which showed a significantly higher BOLD signal after re-learning the task (left, $x = -50, y = -20, z = 48, t_{(31)} = 5.88, p_{FWE-SVC} = 0.002$; right, $x = 30, y = -32, z = 60, t_{(31)} = 5.60, p_{FWE-SVC} = 0.005$); we did not find a comparable effect in IOFC ($p_{FWE-SVC} > 0.05$, Fig. 3b).' (Lines 177-186)

4. Response changes across phases of learning: it is not clear what is actually computed in Figure 3. Is it a beta of GLM, or raw BOLD? Prediction error signals are presumably larger in LN and RN phases, which the authors show to be encoded in OFC. So OFC is just showing prediction error response?

We apologize for the confusing presentation of our results in Figure 3.

Panel c in Figure 3 shows the averaged activation in IOFC and S1 across all trials (HIT, CR, FA, MISS) for each of the four different phases of learning relative to the baseline activity in both regions. The y-axis represents the z-transformed beta value from S1 and IOFC for the different learning phases. Units of the y-axis in the original Figure 3e were arbitrarily chosen, so that all values became positive. We revised the figure in this respect. The updated version now presents the raw activation values after z-transformation (see Fig. R9). The pattern of IOFC and S1 activity across the four phases of learning do not differ when compared to the original results. However, controlling for the outcomes with the updated models, caused smaller SEMs.

Fig. R9 (Updated Fig. 3c in the main text): Averaged activity in IOFC and S1 across all trial types (HIT, CR, FA, MISS) for each of the four learning phases (LN, LE, RN, RE). The error bars indicate the SEM.

Even though we have included reward/error as additional regressors in the univariate GLM analyses, we applied an additional test questioning whether the activity in IOFC was independent of the reward/error outcomes. To this end, we additionally plotted the activity in IOFC for each of the four learning phases separately for HIT, CR, FA and MISS trials. The results are shown in Fig.

R1. We found comparable IOFC activity pattern for each trial type which further supports the argument that IOFC activity was not specifically related to reward or error outcomes (within-subject repeated measures ANOVA, no significant interaction between phases and types of trials ($F(1,31) = 0.95, p = 0.42$) and main effect of trials ($F(1,31) = 0.74, p = 0.53$), only a significant main effect of phases ($F(1,31) = 17.87, p = 0.00019$)). For further information, please also refer to our reply to comment 4 by Reviewer 1.

Based on the updated findings, we can conclude that the activity in IOFC is specifically related to deviations from expected outcome values (prediction error), rather than the outcomes per se (reward, error). These new results are now presented in the updated Fig. 2 and Fig. 3, as well as in the new Supplementary Fig.3.

5. RSA: I am not sure why the data's RDMs are not shown in the paper. It would be more straightforward to see data.

We agree with the reviewer. We re-organized Figure 4 in the main text and added the averaged data RDMs across all participants (Fig. R10). To avoid an overload of information, we now only present the significant results in the updated Figure 4 in the main text. The insignificant results were moved to the Supplementary Fig. 4.

Fig. R10 (Updated Fig. 4 in the main text): The RSA analyses for the stimulus- and outcome-selectivity of response patterns in S1 and lateral OFC, respectively.

6. Stimulus selective vs outcome selective: I was confused about why one expects selective stimulus coding in outcome onsets. Is this related to stimulus value coding? Then why not contrast it to action value coding?

In the RSA analysis, we time-locked the onsets to outcomes to assess stimulus and outcome-selective representations in IOFC and S1. The rationale behind choosing this onset was, that at the time point of outcome presentation, the IOFC should assign credit for unexpected outcomes to specific stimuli (Go-tactile pattern or NoGo-tactile pattern) after reversal of the task rule through signaling the outcome values to the sensory cortex. The sensory cortex, in turn, should remain stimulus-selective when presenting the outcome, but may, at the same time also become outcome-selective due to the feedback from IOFC. Please note that our task is not a pure stimulus-outcome learning task. Instead, it is a stimulus-response-outcome learning task, in which sensory stimuli are directly linked to either a Go or a NoGo action, so that stimulus and action values are redundant to each other.

Motivated by the reviewers' comments, we additionally performed another RSA analysis time-locked to the presentation of the stimulus. With this new RSA analysis, we additionally tested stimulus-selectivity in S1 (Fig. R3). For the corresponding results and discussion, please refer to our reply to comment 5 and 6 from Reviewer 1.

In the revised manuscript, we now explain in more detail why we performed the RSA analysis time-locked to the onset of outcomes and added the result of the new RSA analysis to the supplementary materials.

'To this end, we leveraged a multivariate pattern analysis (i.e., RSA) on the fMRI data at the time of outcome presentation. The rationale behind choosing the presentation of the outcome and not of the stimulus as the onset was, that at the time point of outcome presentation, the IOFC should assign credit for unexpected outcomes to specific stimuli by signaling the outcome values to the sensory cortex. The sensory cortex, in turn, should remain stimulus-selective in the moment of outcome presentation, but may become outcome-selective later in the task due to the ongoing feedback from IOFC. Please note that the stimulus-selectivity includes the associated Go/NoGo response (i.e., stimulus-response selectivity).' (Lines 224-232)

7. PPI analysis. The GLM of PPI analysis does not include important regressors such as outcomes (reward - no reward) or prediction error. All the signal regressors need to be included because otherwise, what is shown here could be a signal correlation instead of a noise correlation.

We thank the reviewer for this suggestion which has helped to extend our PPI model by other confounding factors than those we included in our original model. Similar with the GLM of the univariate analyses, we included the reward/error trials as additional regressors to the new PPI GLM. We did not additionally add PE as a parametric modulator to the GLM for PPI because of the redundancy with the reward/error regressor. The new results are shown in Fig. R11.

As shown in the figure, we found the same significant ipsilateral S1 activity together with the strengthened connectivity to the outcome-selective IOFC immediately following the reversal (RN). We updated the figure and revised the text about S1 activation in the Results section as presented below (**Lines 301-304**).

'We found evidence for a significantly strengthened connectivity immediately after a reversal (RN) between the outcome-selective IOFC subregion and ipsilateral S1 ($x = 20, y = -38, z = 64, t(31) = 4.58, p_{FWE-SVC} = 0.013$, Fig. 6a).'

Fig. R11 (Updated Fig. 6a in the main text): The result of the new PPI analysis. The ipsilateral S1 ($x = 20, y = -38, z = 64$) was shown to have a significantly strengthened connectivity with the outcome-selective IOFC subregion immediately after a reversal (RN>LE).

Reviewer 3 (Remarks to the Author):

In their work, Wang and colleagues study the neural mechanisms by which S1 interacts with prefrontal structures (more specifically IOFC) during a probabilistic tactile learning task. This work can be seen as a follow-up of the original work by one of the co-authors (Banerjee et al 2020), but this time conducted in humans during fMRI, thus seeking integrative consistency across species. While I am enthusiastic about this kind of integrative work, I am less impressed by the specificity of the claims that the authors attempt to make during the presentation of the results both at the level of the modeling and the neural data. This in addition to the lack of clarity in the description of the models and results in several passages of the text. This substantially reduces my enthusiasm for the conclusions stated in this work. I expand on my comments below:

We greatly appreciate the reviewer's critical but constructive points. Following the reviewer's comments, we thoroughly revised the descriptions of the models and results. Briefly, we added 1) comprehensive optimization of parameter values for simulations and model fitting; 2) elaborate descriptions and definitions of parameters in the HGF model; 3) more fMRI analyses and the reason why we specifically investigated IOFC; 4) discussion about the possibility that other prefrontal areas may exert comparable or complimentary interaction with S1 in the context of reversal learning. We hope the reviewer agrees with these changes.

1) Modelling: In several passages of the manuscript, starting from the introduction and keywords, the authors claim that human participants employ a *Bayesian* strategy. Unfortunately, this claim has support neither with their experimental paradigm nor with their modeling approach. First, the behavioral paradigm implemented here has no statistical structure that the participants could learn to claim *Bayesian* belief updating. This is an associative probabilistic task with reversal where no parameter of the environment was systematically manipulated or controlled (e.g., Hazard rates) to be able to apply any form of optimal Bayesian inference.

In this work, the authors employ the HGF to fit their choice data. But employing HGFs and claiming that because it fits the data better, the participants adopt Bayesian inference, is not correct. Note that the RW can also be derived using Bayesian formulations, and in fact, any statistical model can be implemented using Bayesian formulations. But this does not mean that participants invert a Bayesian model based on a learned hierarchical structure of the environment or a given context. This is without mentioning that the HGF employs quite some variational approximations to make the model tractable which then makes the normative approach questionable (e.g., see Piray and Daw 2020). I am not someone who is against Bayesian models (actually quite the opposite), but claims of Bayesian inference need the design of specific behavioral paradigms and the development of the corresponding Bayesian inversion models (e.g., see Ma, 2012 TiCS).

We appreciate the reviewer for pointing out this important issue. We acknowledge that the claims about participants employed a 'Bayesian' strategy was incorrect. We have now removed all interpretations in this context throughout the manuscript. The main difference between the models we applied (mainly RW vs. HGF) is that one model has a fixed "ideal" learning rate across subjects (RW), whereas the other one includes an individual expression of (approximate) Bayes-optimal learning based on the subject-specific learning trajectories (HGF). Based on our findings, we now claim that participants in the reversal learning task did learn the task-relevant conditional probabilities of stimuli and dynamically updated their learning rate. Below we give some examples on how we revised the text:

'Next, we questioned whether participants' learning behavior can be rather explained by hierarchical learning (i.e., Bayesian HGF model), which includes dynamic updates of the learning rate based on individual learning trajectories, or by a fixed "ideal" learning rate as assumed by the reinforcement learning algorithm (RW). To address this question, we applied the random-effect Bayesian model selection (BMS) at the group level, which revealed posterior model probabilities of 95% for the winning HGF model (posterior probabilities: 0.95; exceedance probability = 1.00, Fig. 1g).' (Lines 138-145)

'These results provide evidence that the participants learned the task-relevant conditional probabilities of stimuli and dynamically updated their learning rate in the reversal learning task.' (Lines 149-151)

2) Model specification and fits: The explanation of the model implementation, in particular the HGF is incomplete and difficult to follow without reading the original work. For instance, before equation 8 the authors introduce ϵ , but it is not applied in any of the subsequent equations. Also is ϕ a free parameter? How is δ exactly defined? Additionally, there is no information on how the models are fit to the data. Were the fits being done for each participant? What were the free parameters? (in the case of the HGF, what is with ω ?, etc. or which were the actual free parameters, were some parameters set to fix values, in case yes, then how these were selected).

In the main text line 137, where does the sentence: "particular set of parameters" come from? Moreover, it will be important to show the quality of the model fits. In Figure 1 the only information regarding this is shown in panel e but it is not clear if this corresponds to the average data with the average model predictions of the model fit individually for each participant. It would be more convincing to show how learning behavior as shown in panel d is captured by the model.

Also, please provide a table (main text or supplement) with the average parameters +/- s.d. of the parameters fit for each model that would allow qualitative reproduction of the predictions.

The reviewer brings up two distinct points. First, the incomplete description of the HGF model and second, the lack of information about the model fitting procedure. We will address these points separately.

Explanation of the HGF model. We extensively edited the *Modelling of human behavior* section in the methods, including the definition of all necessary parameters, and all free and fixed parameters. In brief, (1) we applied the two-level version of the perceptual model in HGF, where we eliminated the third level (the log-volatility of the environment) from the hierarchy by fixing both the value of log-volatility ϑ and by setting the couple strength κ between second and third levels to zero. (2) There are two free parameters, one is ω which is a constant component of the step size between consecutive time steps at the second level of the perceptual model. The other is ζ in the response model that captures how deterministically the response is associated with the prior belief. (3) The belief on trial t at the second level is updated based on the outcome prediction error $\delta_1^{(t)}$ weighted by the precision of predictions $\varphi_2^{(t)}$. The outcome prediction error $\delta_1^{(t)}$, which drives learning at the second level of our HGF model, is defined as the difference between the actual outcome and its estimated probability before the outcome is delivered. The precision weight

$\varphi_2^{(t)}$ is updated with every trial and can be regarded as equivalent to a dynamic learning rate in reward learning models.

We now added all these parameter definitions and the corresponding equations to the revised manuscript (**Pages 27-29**).

The choice of parameter values for simulations and model fitting. Please also refer to our reply to comment 1 from Reviewer 2. In brief, we explored the effect of a range of parameter settings to determine how the model-independent measures (p_{correct}) change with different parameter values (Fig. R4). Finally, we chose the parameter values $\beta=5$ for RW and $\zeta=10$ for HGF, based on the simulation results, well capturing the range of parameter values obtained from fitting the real behavioral performance (the averaged fitting parameter values \pm SD for RW and HGF are shown in the following Table R2). The model fitting procedure was conducted with the HGF toolbox using a quasi-Newton minimization algorithm (or BFGS Algorithm introduced by Broyden, Fletcher, Goldfarb, and Shanno) to assess the fitting parameter values for both models.

Table R2 (Supplementary Table 2 in the main text). The fitting parameter values (means and standard errors) for the RW and HGF model

Models	Parameters	Mean	SD
RW	α	0.19	0.03
	β	5.68	3.13
HGF	ω	-3.45	0.59
	ζ	13.48	2.59

For the parameters in the perceptual model (α in RW and ω in HGF), we calculated the Bayes optimal parameter values using the function implemented in the HGF toolbox (see new Table R1 for the ‘optimal’ values; please also refer to comment 1 from Reviewer 2).

Combining the ‘optimal’ perceptual parameter values and the fit response parameters values ($\beta=5$ in RW model and $\zeta=10$ in HGF model), we run simulations and plotted $p(\text{staying})$ as a function of the outcomes (correct or wrong) of the previous trial for each of the four models. Participants’ performance was better fitted by both, the RW and the HGF model. To determine which of these two models is more likely to describe participants’ behavior, we compared relative log-model evidence between the HGF and RW model on the individual level (HGF was superior in 28 out of 32 participants) and group level (posterior probabilities: 0.95; exceedance probability = 1.00). In order to validate the superiority of the winning HGF model, we also plotted the averaged behavioral performance to show how real learning behavior is captured by the simulated responses from the HGF model, as shown in Fig. R12.

Fig. R12 (Supplementary Fig. 2 in the main text). The overlay of real behavioral responses and the simulated responses produced by the winning HGF model. The shaded area indicates the standard error of the mean (SEM).

We have now incorporated model simulations and fitting results in the Supplementary Fig. 1 and Fig.2, and updated Fig.1 in the main text. Additionally, we revised the text to explain how we choose these parameter values (**Lines 116-132**):

'First, we simulated behavior across a range of parameter settings to determine how the model-independent measures change with different parameters (Supplementary Fig. 1). Then, we tested one measure that captures fundamental aspects of the flexible decision process based on prior experience: the probability of repeating a decision, $p(\text{staying})$. To this end, we first estimated the free parameters by fitting them to the real behavioral performance (Supplementary Table 1). The choice of free response parameters (β in RW and ζ in HGF) was based on the simulation results and in the range of parameters obtained from the estimation of model fitting. For the free perceptual parameters (α in RW and ω in HGF), we calculated the 'Bayes optimal' values for a given input sequence μ (Supplementary Table 2). Based on these simulations and model fitting results, we finally simulated the responses and calculated $p(\text{staying})$ using the four different models with a particular set of parameters (M1: $b = 0.5$; M2: $\varepsilon = 0.05$; M3: $\beta = 5$; M4: $\zeta = 0.5$; for more details, please refer to the Methods section). These simulation parameters also match the typical parameter values used in the literature to constrain overall performance (Wilson and Collins, 2019). We repeated the simulation process 1000 times and plotted $p(\text{staying})$ as a function of the outcomes (correct or wrong) of the previous trial for each of the four models (Fig. 1f).'

3) Results section starting on line 184: Please mention what exact parameter of the HGF model

presented in the methods section was used to compute the PEs: was this the choice prediction error? Or is it one of the other higher-level PEs (ϵ) in the HGF? If it is the choice prediction error, it is unclear to me why a well-defined structure playing a role in choice PEs that is present in basically all fMRI studies such as the ventral striatum (including the original work employing HGFs, Iglesias et al, 2013 Neuron) does not show up in this work? If it is not choice prediction error but a higher level prediction error, does this PE match the type of PE studied in Banerjee et al 2020? I presume not as I think in Banerjee et al 2020 the authors studied choice PEs in. A key reason why this is critical to clarify is that both types of PE appear to make different contributions at the neural circuit level (again see: Iglesias et al, 2013 Neuron). In this work, it is also not clear whether both levels of PE from the HGF were introduced as regressors in the same GLM. This is essential as not doing so might bias the results and their interpretation.

We very much appreciate the reviewer's point and apologize for not having been clearer in our previous version of the manuscript. In our study, we used the *outcome prediction error*, $\delta_1^{(t)}$, which is defined as the difference between the actual outcome and its estimated probability before outcome presentation ($\delta_1^{(t)} = u_1^{(t)} - \hat{u}_1^{(t)}$). Only the *outcome prediction error* was introduced as a regressor in the GLM to elucidate the prediction error-related activation in the brain. According to the definition from the study employing HGF (Iglesias 2013), the *outcome prediction error* is different with the *choice prediction error*. The *choice prediction error* is defined as the difference between the correctness of the participant's choice (1 if the choice was correct, 0 otherwise) and the a-priori probability of the participant's choice being correct (according to the running estimate of the Bayesian model). This PE is positive when the participant's choice was correct and negative when it was wrong. Please note that we used an unsigned outcome prediction error (absolute value bounded between 0 and 1), which corresponds to Bayesian surprise. This is equivalent to the prediction error reported in our previous rodent study (Banerjee et al. 2020) where we showed IOFC neurons responded to both unexpected HIT and FA immediately following the reversal.

Iglesias et al. (Neuron, 2013) found that activity in the striatum related to the *choice prediction error*, but not to the precision-weighted prediction error (ϵ_2). The activations Iglesias et al. found in relation to the precision-weighted PE (ϵ_2) are comparable but not identical to our results related to the outcome prediction error $\delta_1^{(t)}$, because ϵ_2 additionally accounted for the precision weight $\varphi_2^{(t)}$. It is also worth mentioning, that Iglesias et al. studied different levels of the precision-weighted PE (ϵ_2 and ϵ_3). In our study, as we mentioned earlier in the response to comment 2, we applied a two-level version of the HGF, where we eliminated the log-volatility of the environment on the third level from the hierarchy by fixing both, the value of log-volatility ϑ and the couple strength κ between second and third levels, to zero. The two-level version of the HGF model assumes a low or stable volatility over the time course of the experiment, which is in line with our experimental settings. In our revised supplement, we now provide all relevant details about the whole-brain fMRI results related to the outcome prediction error (Table R3). These areas may have also contributed to interactions with the sensory cortex which we will continue to explore in the future (please also see our response to comment 6 below).

Table R3 (Supplementary Table 3 in the main text). Brain regions positively related to the outcome prediction error ($p < 0.001$, uncorrected)

Regions	Hemisphere	Peak coordinates			T-score
		x	y	z	
Insula	R	30	22	2	6.02
Insula	L	-28	22	-2	5.63
Posterior parietal cortex	R	34	-64	44	5.82
Orbitofrontal cortex	R	40	48	-2	4.94
Middle frontal gyrus (inf)	R	46	14	32	6.30
Middle frontal gyrus (sup)	R	48	12	50	4.90
Supplementary motor area	R&L	-4	26	42	5.59

We found two distinct clusters in middle frontal gyrus: “inf” indicates its inferior part and “sup” indicates its superior part.

In our revised manuscript, we added more details to better introduce the outcome prediction error. In the Results and Methods sections, we now state the following:

‘We used the winning HGF model, which accounted well for participants’ choice behavior, to derive trial-by-trial estimates of outcome PE ($\delta_1^{(t)}$) at the second level. The outcome PE is the difference between the actual outcome and its a priori probability (i.e., before response outcome observation) according to the model. The unsigned outcome PE (i.e., absolute value) was included into the GLM as a parametric modulator of finite impulse response function (FIR) time-locked to the onset of outcomes and regressed against the fMRI responses in each voxel.’ (Lines 154-160)

‘In this study, we used the unsigned outcome prediction error (i.e., absolute value) which corresponds to Bayesian surprise (den Ouden et al., 2009; Ide et al. 2013), and which is equivalent to the prediction error we investigated in our previous rodent study (Banerjee et al., 2020).’ (Lines 590-592)

4) Results Figure 2: From the description of the main text and the figure legend, it is unclear whether the brain “activations” and activities (in panel c) have any relation to the HGF model or they are mere “activations” relative to baseline. If they are baseline relative activations, it is unclear why for LN trials S1 would have activity with values near zero. This is a sensorial area and it is not expected to have activations at around baseline for the first trials.

We think that the reviewer referred to Figure 3, rather than Figure 2. In our original manuscript, we did not clearly explain the meaning of the y-axis units. Panel c in Figure 3 had no relation to the parameters derived from the HGF model. They represent activation levels in S1 and IOFC relative to baseline for each of the four phases of learning. Please note that this analysis is time-locked to the outcome, not the stimulation delivery and the y-axis presents the z-transformed beta

values extracted from S1 and IOFC. The units represented by the y-axis in the original Figure 3e were arbitrarily chosen, so that all values became positive. Since this caused confusions, we replaced those values by the raw activation z-scores and updated Figure 3e accordingly. Please also refer to our responses to comment 4 from Reviewer 1 and comment 4 from Reviewer 2 for a detailed discussion about this result and some additional analyses we performed (Fig. R7 and R8). In the revised Figure 3 caption, we now state:

'The BOLD signals relative to baseline in IOFC and S1 across the four learning phases (LN, LE, RN, RE). The y-axis indicates the z-score of mean beta value from IOFC and S1 as derived from the general linear model. The error bars indicate the SEM.'

Moreover, this finding goes against the common observation of BOLD adaptation over time, and this has been particularly evidenced in sensory areas. How do the authors explain this discrepancy?

We thank the reviewer for raising this interesting point. In the revised manuscript, we discuss this important issue. Sensory adaptation expresses as a gradual decrease in the responsiveness of the sensory area over time, if the same sensory stimulus is applied to the same skin territory. However, pin indentations in our task changed over trials, and over blocks, and most importantly, participants had to consciously discriminate the tactile pattern and, hence, direct their attention towards the changing pin indentations in order to make a correct decision. Based on these task demands and stimulus features, it is reasonable that S1 did not show BOLD adaptations over time. S1 was rather steadily involved across all phases of learning. We thank the reviewer for raising this point which we now discuss in the revised manuscript (**Lines 212-217**):

'Please note that we applied two different tactile pin indentations over the learning phase in each block and participants had to steadily direct their attention towards these stimulus changes for correct discriminations and decisions. That is why we did not observe any adaptation processes in the form of gradually decreasing S1 responses, which are well known to occur if the same stimulus is repetitively applied to the same skin location over a longer time period (Adibi et al., 2021).'

5) Results Figure 3: It is not entirely convincing that the structure shown in Figure 3a is IOFC, it looks more like the anterior insula. Additionally, are the clusters shown in the two brain slices in Figure 3b the same? If the dimensions of the two slices were kept on the same scale, then the clusters shown in the two cuts appear to be from different cortical locations.

We thank the reviewer for raising this important issue. Based on comment 3 from Reviewer 2, we re-analyzed all fMRI models resulting in slightly changing coordinates and significance levels. Based on these findings, we updated Figure 3 in the main text (see Fig. R5 in the response to the comment 3 from Reviewer 2) by adjusting the brain slices for a better presentation of the brain effects and adding the MNI coordinate for a better orientation.

We found two significant IOFC clusters that correlated with reversals. One was located more posteriorly, directly adjacent to the anterior insula. The other cluster was located in the more anterior part of IOFC. Based on the mask created with the Automated Anatomical Labeling (AAL) atlas, both clusters were assigned to the IOFC (Fig. R13). The latter cluster was closer to the

IOFC cluster that was associated with the encoding of the prediction error (Fig. 2) and the outcome-selective IOFC cluster as identified with the RSA analysis (Fig.5). We now report both IOFC clusters in the updated manuscript as follows (**Line 177-182**):

'First, by comparing LE and RN trials, we observed significantly enhanced BOLD signals in right IOFC immediately after switching the stimulus-response association (two clusters: $x = 38, y = 24, z = -8, t_{(31)} = 7.17, p_{FWE-SVC} < 0.001$; $x = 36, y = 50, z = 0, t_{(31)} = 5.15, p_{FWE-SVC} = 0.02$, Fig. 3a). Based on the Automated Anatomical Labeling (AAL) atlas, both clusters were assigned to the IOFC, but the latter cluster was closer to the IOFC cluster that was associated with the encoding of the prediction error (Fig. 2)'

Fig. R13. Overlay between the activity in IOFC immediately after the reversal and the IOFC mask as created with the AAL atlas.

6) Discussion and interpretation: It is well taken that the authors intend to build a translational story between the rodent work and this one. However, the question here is whether the S1 results are specific to interactions with IOFC. One of the main reasons I mention this is that there were many other prominent structures such as MFG, PPC, SMA that apparently did not require small volume correction in the first set of analyses. Therefore, the question is whether the authors can assure that the IOFC results are specific to this structure. Moreover, structures such as MFG, PPC or SMA are stronger candidates to modify S1 function due to closer proximity and also likely monosynaptic connections to S1. It was the case in rodents, but here it is important to question whether it has been evidenced that the IOFC areas localized in this study have indeed direct connections to S1? In case not, the motivation and discussion around this precise IOFC to S1 anatomical interaction might be quite a stretch, and even more in light of the finding in this study that more proximal structures appear to also be involved in the mechanisms studied in this task.

We apologize for not having been clearer why we specifically investigated IOFC, and not one of the other brain regions, such as MFG, PPC or SMA, whose activity was also related to the prediction error, but proximity to S1 is most likely much closer. Regarding the significance of these brain regions, the original Fig. 2 presented outcome prediction error-related activity at an uncorrected threshold of $p < 0.001$ (red pattern) and, as an additional overlay, small-volume and FWE corrected activity in IOFC at $p < 0.05$ (yellow pattern). In the corresponding GLM, none of

these regions surpassed FWE-correction at $p = 0.05$ level across the whole brain volume (see updated Fig.2) - only IOFC remained significant due to our strong *a priori* hypothesis and thereby justified small-volume correction. To avoid any confusion caused by the overlay, we updated the figure, which now shows the results of the updated model at $p < 0.001$, uncorrected, for the display purpose only (new Fig.2 and Fig.3), but reported the FWE-SVC results in the main text.

Our *a priori* hypothesis about IOFC was motivated by a series of lesion studies in humans (Hornak et al. 2004) and rodents (Banerjee et al. 2020; Boulougouris, Dalley, and Robbins 2007; McAlonan and Brown 2003) that together accumulated compelling evidence for a specific causal role of IOFC in reversal learning – evidence that was far more convincing than for any other region assumed to underpin reversal learning, including those we found in relation to the prediction error. However, we agree with the reviewer that these regions may have an important comparable or complimentary role in the interaction with S1 in the context of reversal learning.

In the revised version of our manuscript, we strengthened our motivation for our *a priori* hypothesis by the lesion studies mentioned above. Furthermore, we give new perspectives on future research and possible alternative candidate regions that are worth being investigated.

'We next studied the involvement of two a priori hypothesized brain areas engaged in the task: S1, which is important for tactile discrimination and sensory-outcome association learning, and the IOFC, which is engaged in the assignment of outcome value. Our hypothesis for IOFC is based on a series of lesion studies in humans (Hornak et al. 2004) and pharmacogenetic silencing and lesion experiments in rodents (Banerjee et al. 2020; Boulougouris et al. 2007; McAlonan and Brown 2003) that together accumulated compelling evidence for a specific causal role of IOFC in reversal learning' (Lines 168-174)

'Except for IOFC, other prefrontal areas such as anterior cingulate cortex, ventromedial prefrontal cortex, or integrative brain areas like posterior parietal cortex, may exert comparable or complementary interaction with S1 in the context of reversal learning, which can be further explored in future studies. We hypothesize that frontal feedback may directly engage sensory areas or such interaction can also happen involving additional integrative areas. The involvement of these areas may, however, be species-specific and task contingency dependent.' (Lines 406-412)

7) RSA analyses: The times between stimulus and outcome appear to be extremely short an average of around two seconds or less appears insufficient for the BOLD response to decay sufficiently.

We thank the reviewer for this important comment.

First, the RSA analyses was time-locked to the onset of outcome presentation to test both stimulus and outcome selectivity in IOFC and S1. Second, according to general practice (Josephs and Henson 1999), we included a jitter (500ms-1500ms) in our task design to disentangle stimulus and outcome related responses (see Fig. 1a in the manuscript). This jitter has helped to sufficiently decorrelate the events of interest. Reviewing the design matrix orthogonality in SPM, we found only weak between-events correlations (the value of cosine of angle between stimulus

regressor and the outcome regressors in design matrix: $\text{Mean} \pm \text{SD} = 0.22 \pm 0.05$, outlined by blue rectangle in Fig. R14).

The weak between-events correlations suggest that our design allowed to separately investigate stimulus and outcome related brain responses.

Fig. R14. The values of cosine of angle between stimulus regressor and outcome regressors outlined by blue rectangle show only weak between-events correlations

Reference:

- Banerjee, Abhishek, Giuseppe Parente, Jasper Teutsch, Christopher Lewis, Fabian F. Voigt, and Fritjof Helmchen. 2020. 'Value-Guided Remapping of Sensory Cortex by Lateral Orbitofrontal Cortex'. *Nature* 585(7824):245–50. doi: 10.1038/s41586-020-2704-z.
- Boulougouris, Vasileios, Jeffrey W. Dalley, and Trevor W. Robbins. 2007. 'Effects of Orbitofrontal, Infralimbic and Prelimbic Cortical Lesions on Serial Spatial Reversal Learning in the Rat.' *Behavioural Brain Research* 179(2):219–28. doi: 10.1016/j.bbr.2007.02.005.
- Brosch, Michael, Elena Selezneva, and Henning Scheich. 2011. 'Representation of Reward Feedback in Primate Auditory Cortex'. *Frontiers in Systems Neuroscience* 5:5. doi: 10.3389/fnsys.2011.00005.
- Cavada, Carmen, Teresa Compañy, Jaime Tejedor, Roelf J. Cruz-Rizzolo, and Fernando Reinoso-Suárez. 2000. 'The Anatomical Connections of the Macaque Monkey Orbitofrontal Cortex. A Review'. *Cerebral Cortex* 10(3):220–42. doi: 10.1093/cercor/10.3.220.
- Fellows, Lesley K., and Martha J. Farah. 2003. 'Ventromedial Frontal Cortex Mediates Affective Shifting in Humans: Evidence from a Reversal Learning Paradigm'. *Brain* 126(Pt 8):1830–37. doi: 10.1093/brain/awg180.
- FitzGerald, Thomas H. B., Karl J. Friston, and Raymond J. Dolan. 2013. 'Characterising Reward Outcome Signals in Sensory Cortex'. *NeuroImage* 83:329–34. doi: 10.1016/j.neuroimage.2013.06.061.
- Folloni, Davide, Elsa Fouragnan, Marco K. Wittmann, Lea Roumazeilles, Lev Tankelevitch, Lennart Verhagen, David Attali, Jean-François Aubry, Jerome Sallet, and Matthew F. S. Rushworth. 2021. 'Ultrasound Modulation of Macaque Prefrontal Cortex Selectively Alters Credit Assignment-Related Activity and Behavior.' *Science Advances* 7(51):eabg7700. doi: 10.1126/sciadv.abg7700.
- van Heukelum, Sabrina, Rogier B. Mars, Martin Guthrie, Jan K. Buitelaar, Christian F. Beckmann, Paul H. E. Tiesinga, Brent A. Vogt, Jeffrey C. Glennon, and Martha N. Havenith. 2020. 'Where Is Cingulate Cortex? A Cross-Species View.' *Trends in Neurosciences* 43(5):285–99. doi: 10.1016/j.tins.2020.03.007.
- Hornak, J., J. O'Doherty, J. Bramham, E. T. Rolls, R. G. Morris, P. R. Bullock, and C. E. Polkey. 2004. 'Reward-Related Reversal Learning after Surgical Excisions in Orbito-Frontal or Dorsolateral Prefrontal Cortex in Humans.' *Journal of Cognitive Neuroscience* 16(3):463–78. doi: 10.1162/089892904322926791.
- Ide, Jaime S., Pradeep Shenoy, Angela J. Yu, and Chiang-shan R. Li. 2013. 'Bayesian Prediction and Evaluation in the Anterior Cingulate Cortex.' *The Journal of Neuroscience: The Official Journal of the Society for Neuroscience* 33(5):2039–47. doi: 10.1523/JNEUROSCI.2201-12.2013.
- Iglesias, S. 2013. 'Hierarchical Prediction Errors in Midbrain and Basal Forebrain during Sensory Learning'. *Neuron* 80:519–530. doi: 10.1016/j.neuron.2013.09.009.

- Josephs, O., and R. N. Henson. 1999. 'Event-Related Functional Magnetic Resonance Imaging: Modelling, Inference and Optimization.' *Philosophical Transactions of the Royal Society of London. Series B, Biological Sciences* 354(1387):1215–28. doi: 10.1098/rstb.1999.0475.
- Kringelbach, Morten L., and Edmund T. Rolls. 2004. 'The Functional Neuroanatomy of the Human Orbitofrontal Cortex: Evidence from Neuroimaging and Neuropsychology.' *Progress in Neurobiology* 72(5):341–72. doi: 10.1016/j.pneurobio.2004.03.006.
- Mars, Rogier B., Saad Jbabdi, and Matthew F. S. Rushworth. 2021. 'A Common Space Approach to Comparative Neuroscience.' *Annual Review of Neuroscience* 44:69–86. doi: 10.1146/annurev-neuro-100220-025942.
- McAlonan, Kerry, and Verity J. Brown. 2003. 'Orbital Prefrontal Cortex Mediates Reversal Learning and Not Attentional Set Shifting in the Rat.' *Behavioural Brain Research* 146(1–2):97–103. doi: 10.1016/j.bbr.2003.09.019.
- Murray, Elisabeth A., and Peter H. Rudebeck. 2018. 'Specializations for Reward-Guided Decision-Making in the Primate Ventral Prefrontal Cortex.' *Nature Reviews. Neuroscience* 19(7):404–17. doi: 10.1038/s41583-018-0013-4.
- Neubert, Franz-Xaver, Rogier B. Mars, Jérôme Sallet, and Matthew F. S. Rushworth. 2015. 'Connectivity Reveals Relationship of Brain Areas for Reward-Guided Learning and Decision Making in Human and Monkey Frontal Cortex.' *Proceedings of the National Academy of Sciences of the United States of America* 112(20):E2695-704. doi: 10.1073/pnas.1410767112.
- den Ouden, Hanneke E. M., Karl J. Friston, Nathaniel D. Daw, Anthony R. McIntosh, and Klaas E. Stephan. 2009. 'A Dual Role for Prediction Error in Associative Learning.' *Cerebral Cortex (New York, N.Y. : 1991)* 19(5):1175–85. doi: 10.1093/cercor/bhn161.
- Petrides, Michael, Francesco Tomaiuolo, Edward H. Yeterian, and Deepak N. Pandya. 2012. 'The Prefrontal Cortex: Comparative Architectonic Organization in the Human and the Macaque Monkey Brains.' *Cortex; a Journal Devoted to the Study of the Nervous System and Behavior* 48(1):46–57. doi: 10.1016/j.cortex.2011.07.002.
- Pleger, Burkhard, Felix Blankenburg, Christian C. Ruff, Jon Driver, and Raymond J. Dolan. 2008. 'Reward Facilitates Tactile Judgments and Modulates Hemodynamic Responses in Human Primary Somatosensory Cortex'. *Journal of Neuroscience* 28(33):8161–68. doi: 10.1523/JNEUROSCI.1093-08.2008.
- Pleger, Burkhard, Christian C. Ruff, Felix Blankenburg, Stefan Klöppel, Jon Driver, and Raymond J. Dolan. 2009. 'Influence of Dopaminergically Mediated Reward on Somatosensory Decision-Making'. *PLoS Biology* 7(7):e1000. doi: 10.1371/journal.pbio.1000164.
- Poort, Jasper, Adil G. Khan, Marius Pachitariu, Abdellatif Nemri, Ivana Orsolich, Julija Krupic, Marius Bauza, Maneesh Sahani, Georg B. Keller, Thomas D. Mrsic-Flogel, and Sonja B. Hofer. 2015. 'Learning Enhances Sensory and Multiple Non-Sensory Representations in Primary Visual Cortex'. *Neuron* 86(6):1478–90. doi: 10.1016/j.neuron.2015.05.037.

- Roelfsema, Pieter R., Arjen van Ooyen, and Takeo Watanabe. 2010. 'Perceptual Learning Rules Based on Reinforcers and Attention'. *Trends in Cognitive Sciences* 14(2):64–71. doi: 10.1016/j.tics.2009.11.005.
- Rudebeck, Peter H., and Elisabeth A. Murray. 2014. 'The Orbitofrontal Oracle: Cortical Mechanisms for the Prediction and Evaluation of Specific Behavioral Outcomes'. *Neuron* 84(6):1143–56. doi: 10.1016/j.neuron.2014.10.049.
- Rudebeck, Peter H., Richard C. Saunders, Dawn A. Lundgren, and Elisabeth A. Murray. 2017. 'Specialized Representations of Value in the Orbital and Ventrolateral Prefrontal Cortex: Desirability versus Availability of Outcomes.' *Neuron* 95(5):1208-1220.e5. doi: 10.1016/j.neuron.2017.07.042.
- Rudebeck, Peter H., Richard C. Saunders, Anna T. Prescott, Lily S. Chau, and Elisabeth A. Murray. 2013. 'Prefrontal Mechanisms of Behavioral Flexibility, Emotion Regulation and Value Updating.' *Nature Neuroscience* 16(8):1140–45. doi: 10.1038/nn.3440.
- Rushworth, Matthew F. S., Mary Ann P. Noonan, Erie D. Boorman, Mark E. Walton, and Timothy E. Behrens. 2011. 'Frontal Cortex and Reward-Guided Learning and Decision-Making'. *Neuron* 70(6):1054–69. doi: 10.1016/j.neuron.2011.05.014.
- Tsuchida, Ami, Bradley B. Doll, and Lesley K. Fellows. 2010. 'Beyond Reversal: A Critical Role for Human Orbitofrontal Cortex in Flexible Learning from Probabilistic Feedback'. *Journal of Neuroscience* 30(50):16868–75. doi: 10.1523/JNEUROSCI.1958-10.2010.
- Vossel, S., M. Bauer, C. Mathys, R. A. Adams, R. J. Dolan, K. E. Stephan, and K. J. Friston. 2014. 'Cholinergic Stimulation Enhances Bayesian Belief Updating in the Deployment of Spatial Attention'. *Journal of Neuroscience* 34(47):15735–42. doi: 10.1523/JNEUROSCI.0091-14.2014.
- Vossel, Simone, Christoph Mathys, Klaas E. Stephan, and Karl J. Friston. 2015. 'Cortical Coupling Reflects Bayesian Belief Updating in the Deployment of Spatial Attention'. *The Journal of Neuroscience* 35(33):11532–42. doi: 10.1523/jneurosci.1382-15.2015.
- Wang, B. A., and B. Pleger. 2020. 'Confidence in Decision-Making during Probabilistic Tactile Learning Related to Distinct Thalamo-Prefrontal Pathways'. *Cerebral Cortex* 30:4677–4688,. doi: 10.1093/cercor/bhaa073.

REVIEWER COMMENTS

Reviewer #1 (Remarks to the Author):

R1: Wang and colleagues have very carefully addressed all the comments that I raised. They have done this by performing an extensive set of new analyses. In the course of doing these they have now provided a number of new results. Some of these, for example the new stimulus-locked representational similarity analyses (RSAs), have allowed the authors to considerably extend and consolidate their arguments. I have no further concerns and I continue to think that this is an interesting manuscript.

Reviewer #2 (Remarks to the Author):

The authors performed additional analyses, but I am still very concerned and confused about this manuscript. I still don't know if their analyses support the author's conclusion.

1. The authors elaborated on what they did for model fitting, but this method is very unconventional. The parameters are not optimized using standard algorithms such as gradient descent and are evaluated in cross-validation. The model was not evaluated based on an out-of-sample test or a common way penalizing parameter (like BIC). The parameters used to generate fMRI prediction were not optimized for each participant's behavior. I am not entirely sure why the authors did not take these standard procedures. The current method seems very problematic.

2. Controlling outcome signals in fMRI analyses for prediction error. I am still confused about this. I thought that outcome was always reward or no reward for the participants. But the authors convert this reward/no reward to some other variables x . Then this converted outcome is $x = 1$ or 0 ? Then the outcome signal that needs to be controlled should be this $x = 1$ vs 0 . How does this x relate to the author's reward/error? What is exactly the error, given that this is a probabilistic reward task?

Reviewer #3 (Remarks to the Author):

In the revised manuscript, the authors have addressed most of my comments. However, I still have some remaining comments related to the following response points.

1-2) Thank you for your clarifications regarding the origin of the PE signals that were used for the analyses. It is still confusing that the authors keep referring to this signal as “outcome PE” (e.g., see figure 2 legend and also in other parts of the text). Why do the authors simply not call it “unexpected reward outcome” as it is defined in the abstract or outcome “surprise”? Otherwise, the use of |PE| gets convoluted and leads to confusion, as effectively the teaching signals are based on surprise and not based on the classical definition of PE (which are actually the signals that are strongly involved in “teaching” how to adjust expected values in RL).

6) The authors suggest that because some structures such as the MFG do not survive FWE-correction (I presume voxel-wise which is not a representative statistic which assumes voxel independency), these structures are not likely candidates and hence not as likely candidates given their prior hypothesis. First, I would be surprised that given the size of the MFG cluster at $p < 0.001$ it would not survive (Montecarlo) cluster correction at the whole brain level (also not sure about the other clusters). Is this the case? I think that the authors have the possibility of exploring to what degree the effects of OFC are specific given that this is an fMRI study, and claims about OFC top-down feedback to S1 could be further strengthened, as the abstract gives the impression that these top-down signals are specific to OFC.

7) Did the authors orthogonalize the regressors in the design matrix? I did not find information about this. This is critical as orthogonalizing does not directly take care of the influence of other potentially correlated factors and all the analyses presented here should be repeated. If orthogonalization was indeed switched off in SPM (see Correction by Iglesias et al 2019 to the 2013 Neuron paper), then this should be clearly stated in the methods section (unless I missed it).

Responses to Reviewers' Comments:

We would like to thank the editor and reviewers for reviewing our revisions and making further comments on the manuscript. We carefully considered these comments and performed additional analyses. We believe that our new findings further strengthened our conclusions. We hope that the reviewers find our revisions appropriate and satisfying.

Below, please find our point-by-point responses to the reviewer's comments. Sentences in black correspond to the reviewer's comments, followed by our responses in blue. We have highlighted the changes we made in the manuscript.

Reviewer 2 (Remarks to the Author):

The authors performed additional analyses, but I am still very concerned and confused about this manuscript. I still don't know if their analyses support the author's conclusion.

1. The authors elaborated on what they did for model fitting, but this method is very unconventional. The parameters are not optimized using standard algorithms such as gradient descent and are evaluated in cross-validation. The model was not evaluated based on an out-of-sample test or a common way penalizing parameter (like BIC). The parameters used to generate fMRI prediction were not optimized for each participant's behavior. I am not entirely sure why the authors did not take these standard procedures. The current method seems very problematic.

We apologize for the remaining ambiguity of our model fitting and optimization procedures. The feedback from the reviewer can be split into three comments. Below please find our response to each of the three comments:

Comment 1: The parameters are not optimized using gradient descent. We agree that gradient descent is a standard optimization strategy used in machine learning and deep learning. But in our case the gradient descent is very computationally expensive. Therefore, we did not apply gradient descent, but the computationally more efficient

Broyden–Fletcher–Goldfarb–Shanno algorithm (BFGS), which is based on Newton's method and an inversion of the Hessian using conjugate gradient techniques. With each iteration, BFGS computes a matrix by which the gradient vector is multiplied to reach into a "better" direction. This is combined with a more sophisticated line search algorithm, to find the "best" value of the step size. For model fitting to the individual observed responses, we used this efficient quasi-Newton minimization algorithm (i.e., BFGS) to assess the maximum a posteriori estimates of the free parameters for both models (the averaged fitting parameter values \pm SD for the RW and HGF model are shown in the following **Table R1**).

Table R1 (Supplementary Table 2 in the main text). The maximum a posteriori estimates of the free parameters (means and standard errors) for the RW and HGF model

Models	Parameters	Mean	SD
RW	α	0.19	0.03
	β	5.68	3.13
HGF	ω	-3.45	0.59
	ζ	13.48	2.59

We added the information about the quasi-Newton minimization algorithm and emphasized its' advantageous computational efficiency in the manuscript as follows:

'Next, we fitted two alternative models (RW and HGF) to the participant's actual behavioral responses to separately estimate the maximum a posteriori estimates of the free parameters. The fitting procedure started at a random initialization of the optimal value and proceeded iteratively as the value converged to the solution based on a computationally efficient quasi-Newton minimization algorithm (i.e., the Broyden–Fletcher–Goldfarb–Shanno or BFGS algorithm) implemented in the HGF toolbox.' (**Lines 646-651**)

Comment 2: The model was not evaluated based on an out-of-sample test or BIC. As the reviewer suggested, one of the important steps to validate models is to perform cross-validation by measuring fit on held-out data. Therefore, we performed an additional validation analysis for the winning HGF model. We fitted the HGF model to the observed behavioral responses in two out of the three runs (training data, e.g., runs 1 and 2) to

estimate the free parameters (ω and ζ). Then, we used these estimated parameters to predict the responses in the remaining “held-out” run (test data, e.g., run 3). The prediction accuracy was calculated, which measures to what extent the predictions for a given set of parameters from the training data correspond to the ‘real’ behavioral responses from the test data. We found a significantly high accuracy when applying the parameters from training data to predict the responses in the test data (t-test vs. 50%, $p < 0.001$, **Fig. R1**). We repeated the same procedures for the other two combinations (i.e., runs 1&3 as training, run2 as test; runs2&3 as training, run1 as test), which showed the same results. These findings suggest that the winning HGF model can be generalized across different data samples.

Fig. R1 (Supplementary Fig. 3 in the main text): The cross-validation test for the HGF model. The parameters (ω and ζ) of the HGF model were estimated by fitting to the real behavioral responses in the training data (i.e., run1 and run2). These parameters were then used to predict behavior in the test data (i.e., run3). The prediction accuracy was calculated, which revealed the generalizability of the model across different data samples. Box plots indicate median (middle line), 25th, 75th percentile (box) and the maximum and minimum (whiskers) as well as outlier (red cross). The dashed red line indicates chance level.

As suggested by the reviewer, there are other methods for model comparison that have an explicit penalty for free parameters, for example, the Bayes Information Criterion (BIC) and Akaike information criterion (AIC). Considering that the BIC is proportional to AIC and penalizes model complexity more heavily, we calculated the BIC based on the maximum

likelihood estimate of the parameters. The BIC revealed that the HGF is superior to the RW model, which further supports our original BMS approach (**Fig. R2**).

Fig. R2 (Supplementary Fig. 2 in the main text). The Bayesian Information Criterion (BIC) for both, the HGF and RW model. The smaller the value, the better the model fit. Each dot represents a single participant.

Please note that, unlike the maximum likelihood-based BIC, the Bayesian Model Selection (BMS) is based on the log-model evidence (LME). BMS is a standard approach in computational neuroimaging (Penny et al. 2004; Stephan et al. 2009) which assesses the relative plausibility of competing models based on their log evidence that represents the negative surprise about the data given the model and quantifies the trade-off between the accuracy (fit) and complexity of a model. The log-model evidence (LME) is calculated as the negative variational free energy under the Laplace assumption. Using BMS/LME, many previous studies have shown that the hierarchical Bayesian model (HGF) explains participants' behaviors in probabilistic associative learning tasks (Iglesias et al. 2013; Lawson et al. 2017; Powers et al. 2017; Vossel et al. 2015) and reversal learning tasks (Suthaharan et al. 2021; Weilhhammer et al. 2018) better than the classical reinforcement learning model. Based on these previous studies, we stick to the LME/BMS to assess model goodness and model comparison, but we added the cross-validation and BIC

results as additional evidence for model optimization. We added these additional results to the main text which reads as follows:

*'In addition to the LME, we calculated the Bayesian information criterion (BIC), which confirmed that the HGF was superior to the RW model (**Supplementary Fig. 2**). To validate the winning HGF model, we additionally performed cross-validations by predicting the responses in held-out data (**Supplementary Fig. 3**) and ensured that the HGF model successfully captured the real behavior of the whole experiment using the optimized parameters from model fitting (**Supplementary Fig. 4**).'* (Lines 150-155)

In the updated manuscript, we added additional details about the LME and BMS for model comparison, as follows:

'By fitting the model to the participants' actual behavioral responses, the measures of model goodness (i.e., log-model evidence, LME) were calculated. The LME is calculated as the negative variational free energy under the Laplace assumption. LMEs can be used to calculate Bayes factors by exponentiating the difference in LME between two models applied to the same dataset. To identify the model that best explained participants' behavior (RW or HGF), we applied Bayesian Model Selection (BMS). BMS is a standard approach in machine learning and computational neuroimaging (MacKay, 1992; Penny et al., 2004) that compares different models to infer on how neurophysiological or behavioral responses were generated. BMS assesses the relative plausibility of competing models based on their log-evidence which represents the negative surprise about the data given the model and quantifies the trade-off between accuracy (fit) and model complexity. We used random effects BMS to account for potential interindividual variability in our dataset and to quantify the posterior probabilities of the two competing models (RW and HGF).' (Lines 651-664)

Comment 3: The parameters used to generate fMRI prediction were not optimized for each participant's behavior. We assume that the reviewer refers to the 'PEs', which we

applied to the fMRI data. By fitting the model to observed responses, the HGF model contains subject-specific parameters that are optimized based on participants' real behavioral responses and hence allows to assess the individual expression of (approximate) Bayes-optimal learning, i.e., the PE trajectories. Therefore, the PEs were calculated for each participant based on their own (subject-specific) best-fitting parameter values. Using the subject-specific PE trajectories, we conducted a trial-by-trial parametric modulation analysis with the BOLD signal to investigate the PE-related signals in IOFC. Therefore, the PE trajectories of a Bayes-optimal observer were indeed optimized and estimated based on the participants' real behavior. We now explicitly describe that the subject-specific PE trajectories were estimated based on participants' real behavioral responses in the **Results** and **Methods** sections:

'We used the winning HGF model, which contains subject-specific parameters optimized based on participants' actual behavioral responses and allows for individual expression of (approximate) Bayes-optimal learning, to drive the subject-specific estimates of outcome PE ($\delta_1^{(t)}$) at the second level.' (**Lines 161-164**)

'Using the subject-specific parameters, optimized based on participants' actual behavioral responses, the individual PE trajectories were finally assessed using the winning HGF model.' (**Lines 664-666**)

We comprehensively revised the corresponding paragraphs on model simulation and fitting in the **Results (Lines 117-129 and lines 140-146)** and added a separate Model simulation and fitting section in the **Methods (Lines 627-666)**.

2. Controlling outcome signals in fMRI analyses for prediction error. I am still confused about this. I thought that outcome was always reward or no reward for the participants. But the authors convert this reward/no reward to some other variables x . Then this converted outcome is $x = 1$ or 0 ? Then the outcome signal that needs to be controlled should be this $x = 1$ vs 0 . How does this x relate to the author's reward/error? What is exactly the error, given that this is a probabilistic reward task?

The reviewer is correct in assuming that the outcome for each trial was always reward or no reward and that there were no errors given the probabilistic nature of our task.

In the last version of our revision, we added the reward/no-reward trials as two additional regressors/variables in the GLM - not $x = 1$ (for reward) or 0 (for no-reward).

Motivated by the reviewers' questions and to better account for the reward/no-reward related variance in the fMRI analyses of the prediction error, we removed the reward/no-reward regressors and added reward/no reward as an additional parametric modulator of interest (1 for reward, 0 for no reward) to the main regressors. For each of the four main regressors (LN, LE, RN, RE), we defined two parametric modulators. The first modulator was the prediction error $\delta_1^{(t)}$, and the second modulator was the reward/no-reward outcome. Please note that we omitted the orthogonalization because we did not impose any judgment on the relative importance of parameters for explaining the fMRI data. Therefore, prediction errors and reward/no-reward outcomes competed to explain the variance.

Using parametric modulation instead of additional variables (regressors) only marginally affected the results of prediction error-related activity (**Fig. R3**).

Fig. R3 (**Fig.2** in the main text). Brain regions related to the outcome PE, including bilateral middle frontal gyrus (MFG), the supplementary motor cortex (SMA), bilateral insular cortex (Ins.), right posterior parietal cortex (PPC), and right lateral orbitofrontal cortex (IOFC). T-maps are displayed at $p < 0.001$, uncorrected, for display purposes only.

To test overall changes of BOLD responses immediately after the reversals (i.e., RN > LE) and during re-learning (i.e., RE > LE), we used a GLM consisting of four main regressors representing the four different phases of the task (LN, LE, RN, RE), without any parametric modulators. We updated the results in **Fig.3** in the main text accordingly (**Fig. R4**).

Fig. R4 (**Fig.3** in the main text). The engagement of IOFC and S1 immediately after reversals and during re-learning, respectively. a. Significantly enhanced BOLD signals in IOFC immediately after the reversal (RN > LE, $p < 0.001$, uncorrected, for display purposes only). b. Significantly enhanced BOLD signals in S1 during re-learning after the reversal (RE > LE, $p < 0.001$, uncorrected, for display purposes only). c. The BOLD signals relative to baseline in IOFC and S1 across the four learning phases (LN, LE, RN, RE). The y-axis indicates the z-score of the mean beta value from IOFC and S1 as derived from the general linear model. The error bars indicate the SEM.

Finally, we want to specifically thank the reviewer for the discussion on the term ‘error trials’ that might have caused confusion because of the probabilistic nature of our task. We agree with the reviewer that there are no ‘real’ error trials as compared to our rodent study (Banerjee et al. 2020). There are only reward and non-reward trials. Therefore, we changed ‘error’ trials to ‘non-reward’ trials throughout the manuscript. We thank the reviewer for this important hint.

Reviewer #3 (Remarks to the Author):

In the revised manuscript, the authors have addressed most of my comments. However, I still have some remaining comments related to the following response points.

1-2) Thank you for your clarifications regarding the origin of the PE signals that were used for the analyses. It is still confusing that the authors keep referring to this signal as “outcome PE” (e.g., see figure 2 legend and also in other parts of the text). Why do the authors simply not call it “unexpected reward outcome” as it is defined in the abstract or outcome “surprise”? Otherwise, the use of $|PE|$ gets convoluted and leads to confusion, as effectively the teaching signals are based on surprise and not based on the classical definition of PE (which are actually the signals that are strongly involved in “teaching” how to adjust expected values in RL).

We thank the reviewer for this comment.

Previous studies emphasize that IOFC responds to both, positive and negative outcomes (Banerjee et al. 2020; O’Doherty et al. 2001), suggesting that IOFC is specialized for assigning credit for rewards *and* errors to specific stimulus choices (Rushworth et al. 2011). Based on these studies, we hypothesized that IOFC encodes deviations from expected outcome values and represents credit assignments for both, positive (unexpected reward) and negative outcomes (unexpected non-reward) after the rule switch. Based on this evidence, we considered all types of trials including reward and non-reward trials (HIT, CR, FA and MISS), and used the term ‘outcome PE’ to refer to unexpected reward and unexpected non-reward outcomes. Based on our results, we

concluded that the representations of these unexpected outcomes (outcome PEs) in IOFC transmit a 'teaching' signal to update the representations in the sensory cortex after a rule switch, irrespective of whether they have been rewarded or not.

In our last revision, as a response to reviewer 1, we tested the hypothesis that responses from IOFC may rather reflect the value of reward *and* no-reward outcomes. We performed additional analyses to separate BOLD activity in IOFC for reward (HIT, CR) and non-reward trials (FA, MISS) across the four phases of the learning task (**Fig. R5**). In agreement with our hypothesis, we found IOFC activity in both, unexpected reward (HIT and CR) and unexpected non-reward trials (FA and MISS) immediately after the reversal, suggesting that the activity in IOFC was not specifically related to reward or non-reward outcomes (within-subject repeated measures ANOVA: no significant interaction between phases and types of trials ($F(1,31) = 0.95, p = 0.42$) and the main effect of types of trials ($F(1,31) = 0.74, p = 0.53$); only a significant main effect of phases ($F(1,31) = 17.87, p = 0.00019$)).

Fig. R5 (Supplementary Fig. 5 in the main text). The BOLD signals in IOFC across the four key learning phases for the four types of trials (HIT, CR, FA, MISS) respectively. The error bars indicate the SEM.

We revised the corresponding paragraph to emphasize the role of IOFC during RN in credit assignment of both, unexpected reward and non-reward outcomes, to specific stimulus-response associations (**Lines 198-210**):

‘During RN, we again found transient but large IOFC responses to unexpected outcomes, which decreased as participants re-learned the task during RE (Fig. 3c). To test the potentially differential influence of appetitive and aversive outcomes, we separately analyzed IOFC activity in rewarded (HIT, CR) and unrewarded trials (FA, MISS) for each of the four learning phases (Supplementary Fig. 5). IOFC responded to both, unexpected reward and

unexpected unrewarded trials immediately after the reversal (RN, within-subject repeated measures ANOVA: no significant interaction between phases and types of trials ($F(1,31) = 0.95, p = 0.42$) and the main effect of types of trials ($F(1,31) = 0.74, p = 0.53$); only a significant main effect of phases ($F(1,31) = 17.87, p = 0.00019$), suggesting that IOFC encodes deviations from expected outcome values after rule reversals to assign credit to specific stimulus-response associations, irrespectively of whether they have been rewarded or not.'

6) The authors suggest that because some structures such as the MFG do not survive FWE-correction (I presume voxel-wise which is not a representative statistic which assumes voxel independency), these structures are not likely candidates and hence not as likely candidates given their prior hypothesis. First, I would be surprised that given the size of the MFG cluster at $p < 0.001$ it would not survive (Montecarlo) cluster correction at the whole brain level (also not sure about the other clusters). Is this the case? I think that the authors have the possibility of exploring to what degree the effects of OFC are specific given that this is an fMRI study, and claims about OFC top-down feedback to S1 could be further strengthened, as the abstract gives the impression that these top-down signals are specific to OFC.

We appreciate the reviewer's comment and apologize that we did not fully address this point in our last revision. We agree with the reviewer that we need to better test the specificity of the IOFC 'teaching' signal to S1.

We made several additional analyses (detailed below) to address this point. We followed the reviewer's suggestion and considered the large frontal MFG cluster, which even presented higher T-values as IOFC, as another plausible candidate region that might have also contributed to S1 teaching signals.

We extracted response pattern from the MFG cluster and performed the same RSA and PPI analyses as for the IOFC. In the RSA analyses, we found that the response patterns in MFG did not significantly represent the stimulus or outcome after reversals (LE → RN). However, during re-learning, the response pattern in MFG was selective for the outcomes (LE → RE, signed-rank

test, $Z_{(31)} = 4.31$, $p < 0.001$, permutation test, effect size = 0.44, $p = 0.03$), but not for the stimulus (Fig. R6). We next tested whether MFG influenced the activity in S1. Using the same PPI analyses as for IOFC, we, however, did not find evidence for a significantly strengthened connectivity between the MFG and S1 immediately after reversals (RN).

Fig. R6 (Supplementary Fig. 8 in the main text). The RSA analyses for MFG. **a.** The representational dissimilarity matrix (RDM) of the stimulus-selective and outcome-selective model. **b.** The RDM of the response pattern in MFG for both LE->RN and LERE. **c.** The comparison between the mean 'similar' (black elements in model RDMs) and mean 'dissimilar' (white elements in model RDMs) using a one-sided Wilcoxon signed-rank test together with permutation tests after reversals (LE->RN) and **d.** same with **c** but during re-learning (LERE). ** indicates $p < 0.01$, *** indicates $p < 0.001$. The results suggest a significant representation of outcomes in MFG only during re-learning (LERE).

These findings further strengthen our conclusion about the specificity of IOFC's involvement - the response pattern in IOFC was outcome-selective after reversals (LE → RN) and during re-learning (LE → → RE), whereas MFG represented outcome-selectivity only during re-learning. More importantly, we did not find a significantly strengthened connectivity between MFG and S1 after reversals. Together, these results provide

evidence that the top-down feedback to S1 after the rule switch originates in IOFC rather than MFG. We added these results as a separate section to the **Results** section and discussed them, as follows:

*'The specificity of IOFC 'teaching' signals to S1. Considering that there were other frontal areas encoding the outcome PE, like MFG, we tested whether the top-down 'teaching' signals to S1 may have alternatively originated from MFG. To this end, we extracted activity from the MFG mask and performed the same RSA and PPI analyses as for IOFC. We specifically tested whether MFG exhibits analogous neural representations and connectivity pattern with S1 as IOFC. In the RSA analyses, we found that the response patterns in MFG did not significantly represent the stimulus or the outcome after reversals (LE→RN, **Supplementary Fig. 8**). However, during re-learning, the response pattern in MFG was selective for the outcomes (LE→→RE, signed-rank test, $Z_{(31)} = 4.31$, $p < 0.001$, permutation test, effect size = 0.44, $p = 0.03$), but not for the stimulus (**Supplementary Fig. 8**). In the PPI analysis, we did not find evidence for a significantly strengthened connectivity between the MFG and S1 immediately after a reversal (RN). Together, these findings provide evidence that the top-down feedback to S1 is specifically related to the IOFC, rather than MFG.'* (**Lines 316- 329**)

*'Except for IOFC, other prefrontal areas such as anterior cingulate cortex, ventromedial prefrontal cortex, or integrative brain areas like posterior parietal cortex, may exert comparable or complementary interaction with S1 in the context of reversal learning, which can be further explored in future studies. Our additional analyses regarding the specificity of IOFC involvement provided an insight into this question. Specifically, the response pattern in the MFG, another frontal area encoding the PE, represented outcome-selectivity only during re-learning and, importantly, was not related to S1 signals, unlike the IOFC (**Supplementary Fig. 8**). Nevertheless we hypothesize that IOFC feedback may directly engage sensory areas, but comparable interactions can also involve other integrative areas. The involvement of these areas, may*

however, be species-specific and task contingency dependent.' (Lines 425-435)

7) Did the authors orthogonalize the regressors in the design matrix? I did not find information about this. This is critical as orthogonalizing does not directly take care of the influence of other potentially correlated factors and all the analyses presented here should be repeated. If orthogonalization was indeed switched off in SPM (see Correction by Iglesias et al 2019 to the 2013 Neuron paper), then this should be clearly stated in the methods section (unless I missed it).

We thank the reviewer for raising this important point and for mentioning the paper by Iglesias et. al. Indeed, they published a correction in 2019 because they incompletely switched off orthogonalization, which affects the interpretation of parametrically modulated regressors.

Due to the comment by the second reviewer, who suggested adding reward/no reward as a second parametric modulator to the main regressors, we are now considering two parameters, the first is the outcome prediction error $\delta_1^{(t)}$, and the second reward/no-reward outcome. We switched off orthogonalization, as recommended, to consider the collinearity of the two parameters. Adding the outcomes as an additional modulator and omitting the orthogonalization only marginally affected the results of prediction error-related activity in IOFC (for more details, please refer to our response to the comment 2 from Reviewer 2 and the updated **Fig. R3**).

We now clearly state that we switched off orthogonalization in the manuscript as follows (Lines 704-708):

'For each of these four main regressors, the absolute value of trial-by-trial outcome prediction error ($\delta_1^{(t)}$) derived from HGF model was defined as a parametric modulator. To control the effect of outcomes on the prediction error signals, another parameter which accounted for the outcomes (i.e., reward/no-reward; 1/0) was added. In this model, we switched off orthogonalization to consider the collinearity of the two parameters.'

References:

- Banerjee, Abhishek, Giuseppe Parente, Jasper Teutsch, Christopher Lewis, Fabian F. Voigt, and Fritjof Helmchen. 2020. 'Value-Guided Remapping of Sensory Cortex by Lateral Orbitofrontal Cortex'. *Nature* 585(7824):245–50. doi: 10.1038/s41586-020-2704-z.
- Iglesias, Sandra, Christoph Mathys, Kay H. Brodersen, Lars Kasper, Marco Piccirelli, Hanneke E. M. DenOuden, and Klaas E. Stephan. 2013. 'Hierarchical Prediction Errors in Midbrain and Basal Forebrain during Sensory Learning'. *Neuron* 80(2):519–30. doi: 10.1016/j.neuron.2013.09.009.
- Lawson, Rebecca P., Christoph Mathys, and Geraint Rees. 2017. 'Adults with Autism Overestimate the Volatility of the Sensory Environment.' *Nature Neuroscience* 20(9):1293–99. doi: 10.1038/nn.4615.
- O'Doherty, J., M. L. Kringelbach, E. T. Rolls, J. Hornak, and C. Andrews. 2001. 'Abstract Reward and Punishment Representations in the Human Orbitofrontal Cortex.' *Nature Neuroscience* 4(1):95–102. doi: 10.1038/82959.
- Penny, W. D., K. E. Stephan, A. Mechelli, and K. J. Friston. 2004. 'Comparing Dynamic Causal Models'. *NeuroImage* 22(3):1157–72. doi: 10.1016/j.neuroimage.2004.03.026.
- Powers, A. R., C. Mathys, and P. R. Corlett. 2017. 'Pavlovian Conditioning-Induced Hallucinations Result from Overweighting of Perceptual Priors.' *Science (New York, N.Y.)* 357(6351):596–600. doi: 10.1126/science.aan3458.
- Rushworth, Matthew F. S., Mary Ann P. Noonan, Erie D. Boorman, Mark E. Walton, and Timothy E. Behrens. 2011. 'Frontal Cortex and Reward-Guided Learning and Decision-Making'. *Neuron* 70(6):1054–69. doi: 10.1016/j.neuron.2011.05.014.
- Stephan, Klaas Enno, Will D. Penny, Jean Daunizeau, Rosalyn J. Moran, and Karl J. Friston. 2009. 'Bayesian Model Selection for Group Studies'. *NeuroImage* 46(4):1004–17. doi: 10.1016/j.neuroimage.2009.03.025.Bayesian.
- Suthaharan, Praveen, Erin J. Reed, Pantelis Leptourgos, Joshua G. Kenney, Stefan Uddenberg, Christoph D. Mathys, Leib Litman, Jonathan Robinson, Aaron J. Moss, Jane R. Taylor, Stephanie M. Groman, and Philip R. Corlett. 2021. 'Paranoia and Belief Updating during the COVID-19 Crisis.' *Nature Human Behaviour* 5(9):1190–1202. doi: 10.1038/s41562-021-01176-8.
- Vossel, Simone, Christoph Mathys, Klaas E. Stephan, and Karl J. Friston. 2015. 'Cortical Coupling Reflects Bayesian Belief Updating in the Deployment of Spatial Attention'. *The Journal of Neuroscience* 35(33):11532–42. doi: 10.1523/jneurosci.1382-15.2015.
- Weilhammer, Veith A., Heiner Stuke, Philipp Sterzer, and Katharina Schmack. 2018. 'The Neural Correlates of Hierarchical Predictions for Perceptual Decisions'. *The Journal of Neuroscience* 38(21):5008–21. doi: 10.1523/jneurosci.2901-17.2018.

REVIEWERS' COMMENTS

Reviewer #2 (Remarks to the Author):

The authors addressed my concerns.

Reviewer #3 (Remarks to the Author):

The authors addressed all my remaining comments and I recommend the manuscript for publication.